# Credible Information Subset Decomposition: An End-to-End Multi-fidelity Learning Model by Modeling Label Information

Sihan Wang [1 2]   Wenjie Du [1 2 3 *]   Yang Wang [1 2 3 *]

## Abstract

In the AI4Chemistry scenario, utilizing heterogeneous data at different fidelity levels is a common and core issue. High-fidelity data is accurate but scarce, while low-fidelity data is abundant but biased. Traditional multi-fidelity methods typically identify cross-fidelity biases based on paired samples under different fidelity labels. However, due to the mismatch in dataset input distribution and the complexity of the biases themselves, these methods are difficult to implement in real-world scientific environments. To address this, we propose a trusted information subset decomposition framework that can efficiently utilize multi-fidelity data without requiring paired samples. Multi-fidelity label supervision is decomposed into three complementary subsets: a trusted information subset based on the absolute value of high-fidelity labels; a trusted subset that captures the reliability of the high-fidelity and medium-fidelity label intervals through adaptive constraints; and an ordered trusted subset representing the numerical relationships within the same fidelity level. These subsets are then integrated into a unified end-to-end model, enabling the reasonable utilization of medium- and low-fidelity information. Extensive experiments on various molecular and material property benchmarks demonstrate that our method consistently outperforms state-of-the-art methods. Code is available at: https://Credible-Information-Subset-Decomposition.

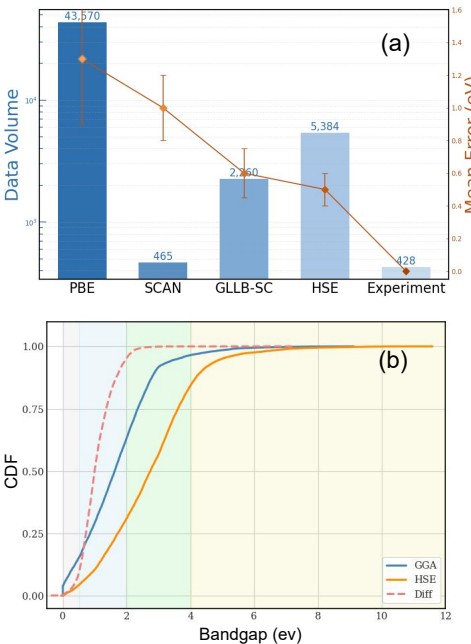

*Figure 1.* (a) Comparison of errors and data volume between high-precision and low-precision calculation methods in bandgap calculation; (b) Compared to HSE, GGA calculation systematically underestimates the bandgap value.

## 1. Introduction

**Background.** In AI-driven chemistry, integrating heterogeneous data to train machine learning models has become a key strategy for accelerating scientific discovery and improving experimental efficiency (Ramakrishnan et al., 2014; Smith et al., 2017; Vermeire & Green, 2021; Butler et al., 2018; Vepreva et al.; Du et al., 2023; 2025b; Chen et al., 2026). Such data often exhibit multi-fidelity characteristics: as dataset (Chen et al., 2021) shown in Fig. 1 (a), there is a trade-off between prediction error and data volume for different fidelity calculation methods. Fig. 1 (b) uses GGA and HSE calculations as examples to demonstrate systematic errors in low precision calculations. In summary, high fidelity data from experiments or high-precision simulations provide high accuracy, but are costly and limited in scale, while low fidelity data are abundant and cost-effective, but have lower accuracy (Yu et al., 2023). Effectively leveraging

[1]University of Science and Technology of China, China [2]Suzhou Institute for Advanced Research, USTC, China [3]State Key Laboratory of Precision and Intelligent Chemistry, USTC, China. Correspondence to: Wenjie Du <duwenjie@ustc.edu.cn>, Yang Wang <angyan@ustc.edu.cn>.

*Proceedings of the 43rd International Conference on Machine Learning*, Seoul, South Korea. PMLR 306, 2026. Copyright 2026 by the author(s).

the complementary strengths of these data sources remains a central challenge in tasks such as molecular design and materials discovery (Batra et al., 2019; Shoghi et al., 2024).

**Related research.** Most existing multi-fidelity methods operate under a paired-sample setting, such as $\Delta$-learning (Ramakrishnan et al., 2015; Atz et al., 2022). These approaches assume that the same input is evaluated at multiple fidelity levels, thereby explicitly modeling systematic computational bias (Fernández-Godino, 2016; Vinod & Zaspel, 2025). However, when encountering unseen samples that lack low-fidelity annotations—such as newly synthesized molecules—this data-coupling assumption often limits their applicability in inductive prediction and real-world scientific discovery scenarios (Pan & Yang, 2009; Buterez et al., 2024). To relax the requirement for cross-fidelity paired data, deep learning methods implicitly assume a shared labeling function or a learnable cross-domain mapping, and attempt to leverage techniques such as domain adaptation or transfer learning (Du et al., 2025a; Zhang et al., 2025; Du et al.). Nevertheless, in scientific multi-fidelity settings, approximation errors are typically heteroscedastic and source-dependent, reflecting systematic biases induced by specific computational methods or experimental protocols rather than random noise (Buterez et al., 2024). Moreover, strategies such as active learning, multi-fidelity Bayesian optimization (Sabanza-Gil et al., 2025; McDonald et al., 2025; Suvarna et al., 2024) may also suffer from similar issues.

**Limitations.** We believe that in practical scientific machine learning scenarios, multi-fidelity data do not necessarily improve pointwise prediction accuracy. Most existing methods rely on several strong assumptions, such as paired samples, simple and identifiable low-fidelity biases, or overlapping input distributions across different fidelities, assumptions that rarely hold in practice (Perdikaris et al., 2017; Li et al., 2020). Many current scientific datasets integrate multi-fidelity measurements with complex and hard-to-identify systematic biases. In particular, many AI4Chemistry datasets provide only input–label pairs and lack intermediate physical information related to fidelity (Yi et al., 2024; Li et al., 2021). Under these circumstances, directly treating low-fidelity labels as approximate substitutes for high-fidelity targets or enforcing cross-fidelity alignment is often unreasonable (Han et al., 2018; Zhang et al., 2024; Goodlett et al., 2023). On the contrary, multi-fidelity data should be regarded as providing heterogeneous supervisory signals with varying levels of reliability. For example, although the absolute label values may be inaccurate, their relative ordering or local consistency can still be reliable. This redefinition requires a re-examination of existing multi-fidelity paradigms: shifting multi-fidelity learning from numerical bias correction toward selectively extracting trustworthy supervisory information, without assuming label comparability, paired samples, or explicit bias models.

**Method.** Based on the above analysis, we re-examines the availability of supervisory signals in multi-fidelity regression and proposes a credible information subset decomposition learning framework for multi-fidelity tasks. The core idea is that multi-fidelity labels are not credible at a unified semantic level; their informational value should be utilized in a structured and hierarchical manner, rather than being uniformly applied to a single point-value regression objective. Specifically, we decompose multi-fidelity supervisory signals into three complementary subsets with different credible semantics: **(i)** an absolute value credible subset, mainly corresponding to high-fidelity data, which predicts the mean and uncertainty through joint modeling of evidence regression; **(ii)** an interval credible subset, which characterizes the reliability of mid-fidelity labels at the scale or error range level, and modulates the prediction uncertainty through adaptive interval constraints; and **(iii)** an ordering credible subset, which utilizes the more stable relative ordering relationships across fidelity data, guiding representation learning through ordering consistency loss. At the model level, we construct a shared latent space through variational representation learning and collaboratively integrate three types of credible supervision within a unified end-to-end optimization framework. This allows us to effectively utilize low-fidelity information and suppress its systematic noise without relying on sample pairing or cross-fidelity label comparability.

- **A Novel Multi-Fidelity Learning Perspective:** Starting from the credibility of supervised signals, a novel multi-fidelity learning perspective is proposed, treating low-fidelity data as carrying partially credible information rather than precise labels.

- **A Credible Information Subset Learning Framework:** Multi-fidelity supervised signals are categorized into three credible subsets: absolute value, interval, and ranking, and learned collaboratively within an end-to-end framework.

- **Stable Performance Improvements:** Experiments on molecular properties and material bandgap tasks demonstrate that this method significantly outperforms existing multi-fidelity and single-fidelity methods under varying fidelity levels and scenarios.

## 2. Related Work

$\Delta$-**Machine Learning Approach** (Ramakrishnan et al., 2015; Forrester et al., 2007)is a representative early work exploring multi-fidelity learning. Methods such as (Frolov et al., 2025; Atz et al., 2022) model the difference between low-fidelity and high-fidelity predictions, improving the accuracy of high-fidelity predictions by learning an error term; this represents the initial form of the multi-fidelity approach. These methods are highly data efficient but heavily

rely on matching low-fidelity and high-fidelity samples, and struggle to handle complex systematic biases in low-fidelity methods or situations where the input spaces do not overlap.

**Core Multi-Fidelity Machine Learning Methods** (Santner et al., 2003; Cutajar et al., 2019; Egorova et al., 2020; Fare et al., 2022; Wang et al., 2021) treat multi-fidelity data as collaborative observations and jointly model them using multi-output GP or auto-regressive GP models. These methods offer strong interpretability, but suffer from high computational complexity with high-dimensional inputs or large-scale data, and have limited ability to model nonlinear complex biases.

With the rise of **deep learning** (Meng & Karniadakis, 2020; Zhang et al., 2024), some works have attempted to combine it with multi-fidelity scenarios, exploring a series of methods based on fidelity label feature enhancement (Chen et al., 2021), transfer learning (Buterez et al., 2024), and multi-objective optimization (Nevolianis et al., 2025). However, these works have limited their focus to large-scale inductive feature learning using low-precision data, and have to some extent neglected the rational use of multi-fidelity labels. In addition, multi-label learning may also be an effective solution, but this method has not yet been applied to scientific multi-fidelity scenarios (Kou et al., 2024; 2025c; 2026b; Wu et al., 2026).

## 3. Method: Credible Subset Decomposition

### 3.1. Task definition

Let $\mathcal{X} \subset \mathbb{R}^d$ denote the input space. Our goal is to learn the mapping function $f^* : \mathcal{X} \to \mathbb{R}$ from inputs to the high-fidelity target labels $\mathcal{Y}^*$. In multi-fidelity learning, we have access to datasets at different fidelity levels:

$$\mathcal{D}^{(s)} = \{(\mathbf{x}_i, y_i^{(s)})\}_{i=1}^{N_s}, \quad s = 0, 1, \ldots, K, \quad (1)$$

where $s = 0$ corresponds to the highest-fidelity data and $s = K$ to the lowest. Labels from lower-fidelity datasets $y_i^{(s)}$ may not exactly equal $f^*(\mathbf{x}_i)$, but they contain a limited set of credible information that can help approximate the high-fidelity target.

From an information-theoretic perspective, multi-fidelity learning aims to leverage the information contained in all fidelity levels to minimize uncertainty about $f^*(x)$.

Let $\mathcal{I}$ denote the complete information of the high-fidelity label. For each fidelity level $s$, we define an associated information subset $\mathcal{I}^{(s)} \subseteq \mathcal{I}^{(s-1)}$. The incremental contribution of fidelity level $s$, conditioned on all higher-fidelity information, can be quantified using conditional mutual information:

$$\Delta I^{(s)} := I\big(\mathcal{I}^{(s)}; f^*(x) \mid \mathcal{I}^{(0)}, \ldots, \mathcal{I}^{(s-1)}\big), \quad (2)$$

where $I(\cdot; \cdot \mid \cdot)$ measures the additional information provided by $\mathcal{I}^{(s)}$ beyond what is already known (Cover, 1999). This formulation highlights that multi-fidelity learning is fundamentally about maximizing predictive relevance under hierarchical information constraints.

To formalize what different fidelity levels contribute, we introduce the concept of *credible information subsets*. For each fidelity level $s$, its information subset $\mathcal{I}^{(s)}$ can be decomposed into three types according to the reliability of the information it provides: absolute credibility, interval credibility, and ranking credibility. Specifically, we denote precise measurements as $I_{\text{abs}}^{(s)} \subset \mathcal{I}^{(s)}$, interval information as $I_{\text{int}}^{(s)} \subset \mathcal{I}^{(s)}$, and relative order information as $I_{\text{rank}}^{(s)} \subset \mathcal{I}^{(s)}$, so that

$$\mathcal{I}^{(s)} = I_{\text{abs}}^{(s)} \cup I_{\text{int}}^{(s)} \cup I_{\text{rank}}^{(s)}, \quad \mathcal{I}^{(s)} \subseteq \mathcal{I}^{(s-1)}. \quad (3)$$

This decomposition reflects the hierarchical structure of multi-fidelity data: low-fidelity datasets mainly provide ranking information $I_{\text{rank}}$, medium-fidelity datasets provide ranking and interval information $I_{\text{rank}} \cup I_{\text{int}}$, and high-fidelity datasets provide ranking, interval, and absolute information $I_{\text{rank}} \cup I_{\text{int}} \cup I_{\text{abs}}$. Importantly, including information outside these credible subsets can *harm the model's predictive ability* by introducing noise or misleading constraints. Therefore, explicitly modeling and selecting credible information is crucial for effective multi-fidelity learning.

In multi-fidelity modeling, the difference in fidelity arises from the gradual loss of computational or experimental conditions. A common approach is to represent higher-fidelity outputs as corrected versions of lower-fidelity outputs:

$$f^{(s-1)}(\mathbf{x}) = \rho^{(s-1)}(\mathbf{x})\, f^{(s)}(\mathbf{x}) + \delta^{(s-1)}(\mathbf{x}), \quad (4)$$

where $\rho^{(s-1)}(\mathbf{x})$ is a learnable correlation coefficient and $\delta^{(s-1)}(\mathbf{x})$ captures additional high-fidelity information (Le Gratiet, 2013; Kennedy & O'Hagan, 2001). From the information-theoretic perspective, the multi-fidelity learning problem can be formalized as minimizing the uncertainty of the high-fidelity prediction conditioned on all credible fidelity information:

$$\hat{f} = \arg\min_{f \in \mathcal{F}} H\big(f^*(x) \mid f(x), \mathcal{I}^{(0)}, \ldots, \mathcal{I}^{(K)}\big), \quad (5)$$

where $H(\cdot \mid \cdot)$ denotes conditional entropy. This formulation naturally balances the trade-off between the richness of high-fidelity information and the broad coverage of low-fidelity data, which often provides more samples but with less credible information per sample.

### 3.2. Fidelity-Agnostic Feature Extraction

As illustrated in Section 3.1, multi-fidelity learning aims to leverage heterogeneous information sources while avoiding uncredible constraints introduced by low-fidelity data.

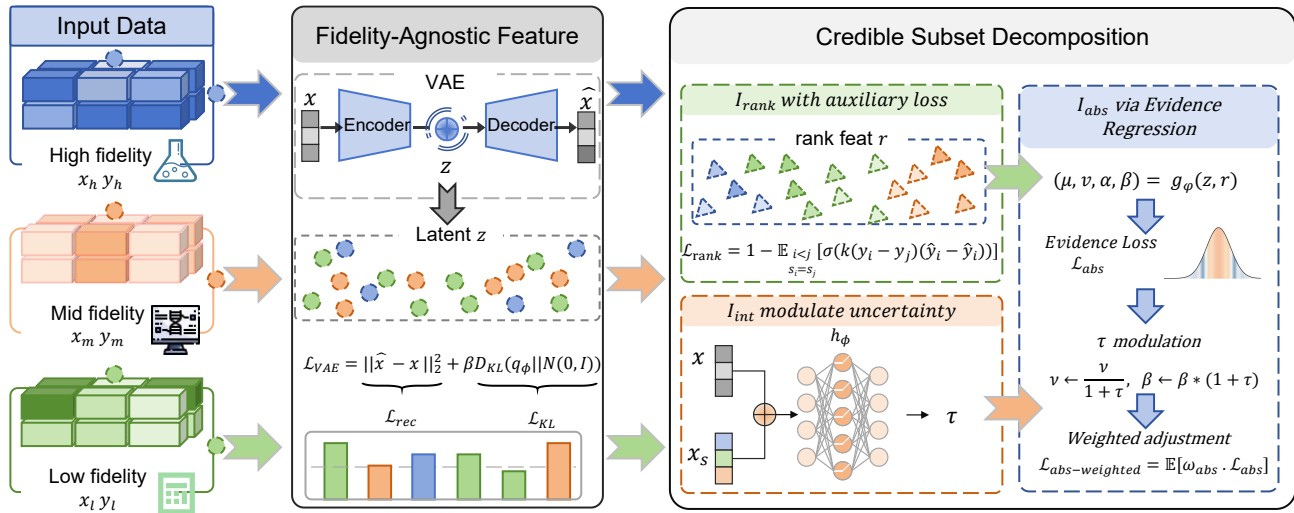

*Figure 2.* End-To-End Framework: for multi-fidelity data with mixed inputs, we first extract fidelity-independent features from the samples, and then decompose the labels into three information subsets based on fidelity, and learn them separately.

Therefore, the key to initial feature extraction lies in preserving the intrinsic structure of the input space $\mathcal{X}$, independent of fidelity-specific labels and supervision. Thus, we employ label-independent variational representations as the first-stage feature extractor. By learning latent variables solely from the input distribution, it captures information shared across fidelity levels while preventing fidelity-induced noise from leaking into the representation.

Let the model input be $\mathbf{x} \in \mathcal{X}$. Introduce a latent variable $\mathbf{z}$ and define the generative model:

$$p_\theta(\mathbf{x}, \mathbf{z}) = p(\mathbf{z})\, p_\theta(\mathbf{x} \mid \mathbf{z}), \qquad (6)$$

with a standard Gaussian prior $p(\mathbf{z}) = \mathcal{N}(\mathbf{0}, \mathbf{I})$. The intractable posterior is approximated by:

$$q_\phi(\mathbf{z} \mid \mathbf{x}) = \mathcal{N}\big(\boldsymbol{\mu}_\phi(\mathbf{x}), \operatorname{diag}(\boldsymbol{\sigma}_\phi^2(\mathbf{x}))\big). \qquad (7)$$

And learn by maximizing ELBO:

$$\mathcal{L}_{\text{ELBO}} = \mathbb{E}_{q_\phi(\mathbf{z}|\mathbf{x})}\big[\log p_\theta(\mathbf{x} \mid \mathbf{z})\big] - D_{\text{KL}}(q_\phi(\mathbf{z} \mid \mathbf{x}) \,\|\, p(\mathbf{z})), \qquad (8)$$

It depends only on the input $\mathbf{x}$, and is independent of the fidelity label. Therefore, $\mathbf{z}$ captures sample structure information that is independent of fidelity.

The encoder takes the global descriptor $\mathbf{x}_{\text{global}}$, original features $\mathbf{x}_{\text{origin}}$ and fidelity descriptor $\mathbf{x}_{\text{s}}$ as input, and outputs the mean and log-variance and he decoder reconstructs the input features from $\mathbf{z}$:

$$(\boldsymbol{\mu}, \log \boldsymbol{\sigma}^2) = \operatorname{Enc}_\phi(x), \ (\hat{\mathbf{x}}_{\text{global}}, \hat{\mathbf{x}}_{\text{origin}}) = \operatorname{Dec}_\theta(\mathbf{z}),$$
$$\mathbf{z} = \boldsymbol{\mu} + \boldsymbol{\sigma} \odot \boldsymbol{\epsilon}, \quad \boldsymbol{\epsilon} \sim \mathcal{N}(\mathbf{0}, \mathbf{I}). \qquad (9)$$

The vae loss is defined as:

$$\mathcal{L}_{\text{VAE}} = \mathcal{L}_{\text{rec}} + \alpha_{kl}\, \mathcal{L}_{\text{KL}}, \qquad (10)$$

where the reconstruction loss $\mathcal{L}_{\text{rec}} = \left\|\hat{\mathbf{x}} - \mathbf{x}\right\|_2^2$ and the KL divergence $\mathcal{L}_{\text{KL}} = D_{\text{KL}}(q_\phi(\mathbf{z} \mid \mathbf{x}) \,\|\, \mathcal{N}(\mathbf{0}, \mathbf{I}))$. The weighting coefficient $\alpha_{kl} = 1e-3$ is used to avoid over-regularization.

### 3.3. Learn $I_{\text{abs}}$ via Evidence Regression

Absolute credible information $I_{\text{abs}}$ corresponds to credible point-wise measurements that directly constrain the true highest fidelity target value $f^*(x)$. This type of information has a certainty that varies depending on the sample and the fidelity. Therefore, learning $I_{\text{abs}}$ requires explicitly modeling the cognitive uncertainty of the prediction. Traditional multi-fidelity regression tasks treat all labels as having the same reliability and learn based on mean squared error (MSE) (Chen et al., 2021; Buterez et al., 2024):

$$\mathcal{L}_{\text{MSE}} = \mathbb{E}_{(x,y)}\left[\|f(x) - y\|_2^2\right]. \qquad (11)$$

In multi-fidelity scenarios, absolute labels from different fidelity sources may have varying degrees of uncertainty. Enforcing these uncertainties uniformly through MSE could introduce misleading constraints, violating the principle of using only a subset of credible information. In contrast, evidence regression models the distribution of the prediction distribution, enabling the learner to simultaneously represent the prediction and its associated uncertainty (Northcutt et al., 2021; Amini et al., 2020; Zhang et al., 2024).

Given the latent representation $z$, the evidential regressor predicts four parameters:

$$(\mu, v, \alpha, \beta) = g_\psi(z), \qquad (12)$$

where $\mu$ is the predictive mean, and $(v, \alpha, \beta)$ parameterize the predictive uncertainty. Positivity constraints are enforced via `softplus` (Zheng et al., 2015).

The loss combines a Student-$t$ negative log-likelihood and an evidence regularization term. To account for multi-fidelity, we weight each sample according to its absolute credibility:

$$\mathcal{L}_{\text{abs-weighted}} = \mathbb{E}\big[w_{\text{abs}} \cdot \mathcal{L}_{\text{abs}}\big], \quad w_{\text{abs}} = \frac{1}{1 + 2s}$$
$$where \quad \mathcal{L}_{\text{abs}} = \text{NLL}(\mu, v, \alpha, \beta; y) + \lambda \, |y - \mu| \, (2v + 2\alpha). \tag{13}$$

### 3.4. Modulate Uncertainty to Capture $I_{\text{int}}$

Low-fidelity or medium-fidelity labels typically provide interval information. Simulation results might indicate that the real target lies within a tolerance range, or experimental measurements might exhibit inherent variability. Unlike $I_{\text{abs}}$, $I_{\text{int}}$ relaxes the target within a credible region, thus encoding credible but imprecise knowledge about the target. To capture this variability, we introduce an interval-aware encoder that captures the interval scale $\tau$ of samples from the perspective of uncertainty modulation:

$$\tau = h_\phi(\mathbf{x}_{\text{global}}, \mathbf{x_s}), \tag{14}$$

where $\mathbf{x_s}$ is fidelity descriptor, and $\tau \geq 0$ represents the width of the credible interval. The encoder is implemented as `MLP` using the `ELU` activation function, followed by `Softplus` to enforce positivity.

This interval estimate is then used to modulate the predictive uncertainty of the evidential regressor:

$$v \leftarrow \frac{v}{1 + \tau}, \quad \beta \leftarrow \beta * (1 + \tau), \tag{15}$$

here $v$ and $\beta$ from Eq (12) are variance-related parameters. A larger $\tau$ is more lenient in uncertainty, thus accommodating the fuzzy boundaries provided by low-fidelity, while a smaller $\tau$ tightens the credible region of high-fidelity.

By explicitly modeling $\tau$, the network can adaptively adjust its credible based on the characteristics of interval samples and their fidelity information, thereby maintaining robust uncertainty quantification.

### 3.5. Extract $I_{\text{rank}}$ using auxiliary rank loss

The ranking credible information $I_{\text{rank}}$ contains information related to label variations, attempting to constrain the monotonic structure of $f^*(x)$. In multi-fidelity settings, this information is often preserved even if low-fidelity labels exhibit scale distortion, making ranking constraints an additional robust source of supervisory information.

To separate change-sensitive information from value-dependent features, we introduce a rank-sensitive representation $\mathbf{r}$:

$$\mathbf{r} = \text{RankEnc}(\mathbf{x}_{\text{global}}, \mathbf{x_s}), \tag{16}$$

where the rank encoder further incorporates fidelity-aware modulation and attention reweighting:

$$\text{RankEnc}(\cdot) = \big(\sigma(W\mathbf{x}_{\text{global}}) \odot W_s\mathbf{x_s}\big) \odot \mathbf{a}, \tag{17}$$

The loss function is constructed based on the ranking prediction $\hat{y}_{\text{rank}} = h(\mathbf{r})$ (Burges et al., 2005; Cao et al., 2007):

$$\mathcal{L}_{\text{rank}} = 1 - \mathbb{E}_{\substack{i < j \\ s_i = s_j}} \big[\sigma\big(k(y_i - y_j)(\hat{y}_i - \hat{y}_j)\big)\big], \tag{18}$$

where only sample pairs from the same fidelity level ($s_i = s_j$) are considered to avoid systematic bias across fidelity.

$\mathbf{r}$ is concatenated with the latent variable $\mathbf{z}$ in Eq (12) and used together for evidence prediction:

$$(\mu, v, \alpha, \beta) = g_\psi(z, r). \tag{19}$$

Its main learning signal is controlled by $L_{\text{rank}}$, thus decoupling credibility-aware supervision and enabling robust knowledge transfer from low-fidelity data without introducing misleading numerical constraints.

### 3.6. Overall Training Objective

Based on trusted information decomposition, the framework proposed in this paper jointly learns fidelity-independent structures, absolute value constraints, interval uncertainties, and ranking consistency through a unified optimization objective. All components are trained end-to-end using a weighted multi-objective loss function:

$$\mathcal{L}_{\text{total}} = \alpha_{\text{vae}} \, \mathcal{L}_{\text{VAE}} + \alpha_{\text{abs}} \, \mathcal{L}_{\text{abs-weight}} + \alpha_{\text{rank}} \, \mathcal{L}_{\text{rank}}, \tag{20}$$

where $\alpha_{\text{vae}}$, $\alpha_{\text{abs}}$, and $\alpha_{\text{rank}}$ control the relative contributions of structural representation learning, absolute credible supervision, and ranking credible supervision, respectively.

## 4. Experiments

### 4.1. Experimental framework

To verify the effectiveness of the trusted information subset decomposition framework, this section designs three parts of experiments: In section 4.3, we test on two data sets of different fidelity to evaluate the improvement of the method compared to the baseline and the effect of multi-fidelity fusion on the highest fidelity sample label prediction. In section 4.4, we conduct visual experiments to show the actual role of each part of the framework in the model training and prediction process. In section 4.5, we conduct rigorous ablation and hyperparameter experiments to evaluate the effective contribution of each module to the framework.

## 4.2. Experimental setup

**Datasets.** In the context of material bandgap prediction, we demonstrate the effectiveness of our method across different data volumes and fidelity levels using a two-fidelity and a five-fidelity dataset. The two-fidelity dataset (**TF-Bandgap**) (Kim et al., 2020) contains bandgap data calculated from 10,481 hybrid functional datasets. High-fidelity labels were calculated using HSE, and low-fidelity labels using GGA. A total of 8,787 usable data points were processed; one-tenth were designated as the high-fidelity dataset, and the remainder as the low-fidelity dataset. The five-fidelity dataset (**FF-Bandgap**) (Chen et al., 2021), from highest to lowest fidelity, consists of 43,570 PBE datasets, 465 Scan datasets, 5,384 HSE datasets, 2,260 GLLB-SC datasets, and 428 experimental datasets. Furthermore, we conducted experiments predicting QM7b (Blum & Reymond, 2009; Rupp et al., 2012) related molecular properties in a three-fidelity scenario to demonstrate the method's cross-scenario versatility in the Appendix E.1 .

**Baseline.** In the single-fidelity (**1-fi**) setting (using only the highest fidelity data), we incorporated the baseline models of Random Forest Regression (RFR) (Chen et al., 2021), Multilayer Perceptron (MLP) (Chen et al., 2021) and Graph Isomorphic Neural Network (GIN) (Xu et al., 2018). In the dual-fidelity (**2-fi**) setting, we use standard transfer learning (Bengio, 2012) as the baseline. For more fidelity ($\geq$ **2-fi**) settings, we use multi-fidelity state embedding (MFSE) (Chen et al., 2021), multi-fidelity GNN (MFGNN) (Buterez et al., 2024), and D-MPNN multi-target(DMMT) (Nevolianis et al., 2025) as baselines.

**Evaluation Metrics.** The experimental results include Mean Absolute Error (MAE), Root Mean Square Error (RMSE) and ranking scores Kendall's Tau-b ($\tau_b$). We repeated the training five times and recorded the mean and standard deviation of three metrics.

**Experimental settings.** The model is trained with the Adam optimizer (Kingma, 2014) ($lr = 1e^{-4}$) under batch size $= 128$. The training process was early-stopped if the validation MAE not increasing over 50 epochs, or the maximum training epochs 200 was reached. The code is primarily implemented using PyTorch and runs on an NVIDIA vGPU 32G.

## 4.3. Method performance

This section presents performance evaluation experiments on two material bandgap datasets (TF-Bandgap and FF-Bandgap) under different fidelity settings. Table 1 verifies the effectiveness of the proposed multi-fidelity method compared to traditional domain transformation methods such as direct training and transfer learning, as well as other multi-fidelity baselines. Furthermore, Figure 3 compares

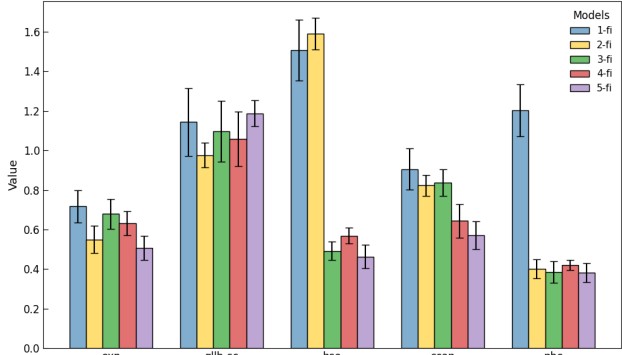

*Figure 3.* The impact of mixing data of different fidelity on test set results. Models 1-fi to 5fi contain training sets {exp}, {exp, pbe}, {exp,hse,pbe}, {exp,hse,pbe,gllb-sc} and {exp,hse,pbe,scan,gllb-sc} respectively. Performance metric reports for MAE.

the impact of training the model with mixed data of different fidelities on the final test results to observe the rationality of the multi-fidelity method itself:

**Obs 1. In TF-Bandgap dataset, credible subset decomposition method outperforms direct training, transfer learning, and other baseline methods.** Our method achieves a MAE of $0.57_{\pm0.012}$, an RMSE of $0.78_{\pm0.018}$, and $\tau_b$ of $0.68_{\pm0.027}$. Compared to the best multi-fidelity baseline (MFSE), the MAE is reduced by approximately 11.7%, the RMSE by approximately 19.5%, and the ranking score is improved by approximately 17.2%. This is because our method avoids the noise interference caused by directly using low-fidelity labels as approximate proxies, while making full use of the effective information at different fidelity, thereby improving the model's cross-domain generalization.

**Obs 2. The credible subset decomposition framework remains robust across different fidelity scenarios.** On the FF-Bandgap dataset, the proposed method maintains leading performance in both two-fidelity and five-fidelity settings. Particularly in the five-fidelity scenario, our MAE reaches $0.50_{\pm0.007}$, RMSE is $0.71_{\pm0.010}$, and $\tau_b$ is $0.70_{\pm0.033}$, significantly outperforming the second best MFGNN whose $MAE = 0.56_{\pm0.008}$, $RMSE = 0.83_{\pm0.013}$ and $\tau_b = 0.68_{\pm0.047}$. This demonstrates that our method can effectively integrate information from multi-fidelity. The hierarchical utilization of credible subsets adapts to the gradual accumulation of cross-fidelity information, making more rational use of multi-fidelity data than methods that rely on feature enhancement (MFSE) or modified transfer learning (MFGNN).

**Obs 3. The mixing and appropriate use of data with different fidelity improved the model's performance.** As shown in Figure 3, the introduction of more high-fidelity data led to a decrease in the overall MAE of the model. However, we also observed that the 4-fi model performed

*Table 1.* Performance comparison on bandgap prediction datasets under different fidelity settings. The best model is indicated in **bold**, and the second-best model is indicated by underline. Performance metrics reports are the mean$_{\pm variance}$.

| Method | TF-Mode | FF-Mode | TF-Bandgap | | | FF-Bandgap | | |
|---|---|---|---|---|---|---|---|---|
| | | | MAE ↓ | RMSE ↓ | $\tau_b$ ↑ | MAE ↓ | RMSE ↓ | $\tau_b$ ↑ |
| **Single fidelity** | | | | | | | | |
| RFR (Chen et al., 2021) | HSE | Exp | $0.82_{\pm0.025}$ | $1.15_{\pm0.033}$ | $0.31_{\pm0.024}$ | $1.41_{\pm0.038}$ | $2.19_{\pm0.051}$ | $0.05_{\pm0.026}$ |
| MLP (Chen et al., 2021) | HSE | Exp | $0.79_{\pm0.018}$ | $1.03_{\pm0.022}$ | $0.38_{\pm0.028}$ | $0.93_{\pm0.026}$ | $1.37_{\pm0.028}$ | $0.14_{\pm0.049}$ |
| GIN (Xu et al., 2018) | HSE | Exp | $0.73_{\pm0.014}$ | $0.92_{\pm0.019}$ | $0.46_{\pm0.035}$ | $0.72_{\pm0.015}$ | $0.88_{\pm0.020}$ | $0.47_{\pm0.037}$ |
| **Dual fidelity** | | | | | | | | |
| Transfer Learning (Bengio, 2012) | GGA+HSE | PBE+Exp | $0.74_{\pm0.018}$ | $1.06_{\pm0.026}$ | $0.47_{\pm0.035}$ | $0.70_{\pm0.013}$ | $0.95_{\pm0.018}$ | $0.51_{\pm0.031}$ |
| MFSE (Chen et al., 2021) | GGA+HSE | PBE+Exp | $\underline{0.64_{\pm0.007}}$ | $\underline{0.97_{\pm0.010}}$ | $\underline{0.58_{\pm0.025}}$ | $\underline{0.61_{\pm0.008}}$ | $0.86_{\pm0.012}$ | $\underline{0.56_{\pm0.019}}$ |
| MFGNN (Buterez et al., 2024) | GGA+HSE | PBE+Exp | $0.68_{\pm0.011}$ | $0.99_{\pm0.017}$ | $0.55_{\pm0.028}$ | $0.63_{\pm0.013}$ | $\underline{0.85_{\pm0.020}}$ | $\underline{0.56_{\pm0.025}}$ |
| DMMT (Nevolianis et al., 2025) | GGA+HSE | PBE+Exp | $0.71_{\pm0.016}$ | $1.02_{\pm0.025}$ | $0.48_{\pm0.037}$ | $0.70_{\pm0.016}$ | $0.93_{\pm0.024}$ | $0.52_{\pm0.035}$ |
| **Credible Subset Decomposition** | GGA+HSE | PBE+Exp | $\mathbf{0.57_{\pm0.012}}$ | $\mathbf{0.78_{\pm0.018}}$ | $\mathbf{0.68_{\pm0.027}}$ | $\mathbf{0.54_{\pm0.009}}$ | $\mathbf{0.72_{\pm0.013}}$ | $\mathbf{0.68_{\pm0.028}}$ |
| **Multi-fidelity** | | | | | | | | |
| MFSE (Chen et al., 2021) | – | 5-fidelity | – | – | – | $0.58_{\pm0.012}$ | $0.84_{\pm0.019}$ | $0.66_{\pm0.035}$ |
| MFGNN (Buterez et al., 2024) | – | 5-fidelity | – | – | – | $\underline{0.56_{\pm0.008}}$ | $\underline{0.83_{\pm0.013}}$ | $\underline{0.68_{\pm0.047}}$ |
| DMMT (Nevolianis et al., 2025) | – | 5-fidelity | – | – | – | $0.62_{\pm0.009}$ | $0.88_{\pm0.015}$ | $0.59_{\pm0.031}$ |
| **Credible Subset Decomposition** | – | 5-fidelity | – | – | – | $\mathbf{0.50_{\pm0.007}}$ | $\mathbf{0.71_{\pm0.010}}$ | $\mathbf{0.70_{\pm0.033}}$ |

slightly worse than the 3-fi model on the HSE, SCAN, and PBE test sets. Analysis revealed that this was due to the significant differences in samples and labels between the gllb-sc data and the other four datasets. Therefore, initial data analysis is essential when training with mixed high-fidelity data. For data with large discrepancies, even with credible subset decomposition within the framework, the model can still be affected by this systematic bias.

### 4.4. Visualization analysis

**Obs 4. On the TF-Bandgap dataset, the interval modulation signal $\tau$ gradually widens and is related to the sample fidelity.** As shown in Figure 4 (a), after approximately 150 rounds, $\tau$ gradually converges to a stable value, and the $\tau$ value corresponding to low-fidelity data (GGA) is significantly larger than the $\tau$ value corresponding to high-fidelity data (HSE). This aligns with the design goal of the interval credible subset $I_{\text{int}}$: a larger $\tau$ value implies a wider credible interval, which can accommodate noise and bias in low-fidelity data; a smaller $\tau$ value tightens the constraint on high-fidelity data, thereby ensuring the accuracy of absolute value prediction.

**Obs 5. The distribution of the ranking-sensitive representation r is affected by fidelity, and the distinction is quite obvious.** As can be seen from Figure 4(b), constrained by Eq. (16) and Eq. (18), the representations **r** of samples with consistent ranking relationships under different fidelity are clustered together, and the cluster shape is significantly different from **z** in Figure 4(c). This indicates that the method does indeed decouple the learning of ranking information other than absolute value from the labels and data, providing

the model with additional credible supervision signals, thus explaining the improvement of $\tau_b$.

**Obs 6. The variational representation features z are not significantly correlated with fidelity.** Features from samples of different fidelity are uniformly distributed in the latent space, without obvious clustering based on fidelity labels. This confirms that the fidelity-independent feature extraction module effectively captures the intrinsic structural information of the input, regardless of the fidelity label. This indicates that the shared latent space **z** avoids low-fidelity noise leakage into the representation, laying a solid foundation for the subsequent integration of multiple credible subsets.

### 4.5. Ablation experiment and hyperparameter analysis

**Ablation experimental setup.** We verified the function of the three modules respectively. (1) w/o $I_{\text{abs}}$: Instead of evidence regression, MSE loss is used, and $\tau$ is directly used to modulate the MSE loss; (2) w/o $I_{\text{int}}$: No longer modulate uncertainty with $tau$; (3) w/o $I_{\text{rank}}$: The learning and concatenation of ranking auxiliary feature **r** and their loss functions will no longer be performed.

**Obs 7. Removing any credible subset will result in a performance degradation.** As shown in Table 2, compared with the vanilla model, removing $I_{\text{abs}}$ significantly reduced MAE prediction performance, indicating the core role of backbone information learning in the framework. Furthermore, removing $I_{\text{abs}}$ also weakens the role of $I_{\text{int}}$, impacting performance to some extent. Removing $I_{\text{rank}}$ primarily leads to a loss of $tau_b$ scores, diminishing the model's inductive and guiding role in real-world experimental scenarios. Over-

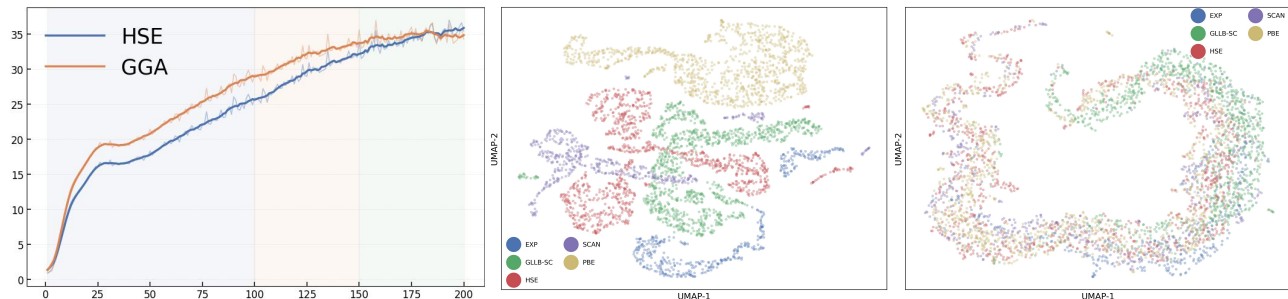

*Figure 4.* (a)The modulation signal $\tau$ of $I_{\text{int}}$ changes with the training epochs; (b) The distribution of order-sensitive representation **r** from Eq. (16) on the FF-Bandgap dataset; (c) Distribution of feature **z** from Eq. (9) on the FF-Bandgap dataset.

*Table 2.* Ablation experiment on bandgap prediction datasets. TF-Bandgap is under 2-fidelity and FF-Bandgap is under 5-fidelity.

| Method | TF-Bandgap | | | FF-Bandgap | | |
|---|---|---|---|---|---|---|
| | MAE ↓ | RMSE ↓ | $\tau_b$ ↑ | MAE ↓ | RMSE ↓ | $\tau_b$ ↑ |
| **w/o $I_{\text{abs}}$** | $0.66_{\pm0.013}$ | $0.85_{\pm0.019}$ | $0.61_{\pm0.021}$ | $0.58_{\pm0.009}$ | $0.80_{\pm0.015}$ | $0.66_{\pm0.027}$ |
| **w/o $I_{\text{int}}$** | $0.60_{\pm0.014}$ | $0.82_{\pm0.018}$ | $0.63_{\pm0.024}$ | $0.55_{\pm0.013}$ | $0.78_{\pm0.020}$ | $0.68_{\pm0.028}$ |
| **w/o $I_{\text{rank}}$** | $0.58_{\pm0.005}$ | $0.81_{\pm0.008}$ | $0.62_{\pm0.035}$ | $0.53_{\pm0.006}$ | $0.75_{\pm0.008}$ | $0.64_{\pm0.032}$ |
| **vanilla model** | $\mathbf{0.57}_{\pm0.012}$ | $\mathbf{0.78}_{\pm0.018}$ | $\mathbf{0.68}_{\pm0.027}$ | $\mathbf{0.50}_{\pm0.007}$ | $\mathbf{0.71}_{\pm0.010}$ | $\mathbf{0.70}_{\pm0.033}$ |

*Table 3.* Hyperparameter study of $\alpha_{\text{abs}}$ and $\alpha_{\text{rank}}$. TF-Bandgap is under 2-fidelity and FF-Bandgap is under 5-fidelity.

| Setting | TF-Bandgap | | | FF-Bandgap | | |
|---|---|---|---|---|---|---|
| | MAE ↓ | RMSE ↓ | $\tau_b$ ↑ | MAE ↓ | RMSE ↓ | $\tau_b$ ↑ |
| **Fixed $\alpha_{\text{rank}} = 5e-3$** | | | | | | |
| $\alpha_{\text{abs}} = 5.0$ | 0.58 | 0.81 | 0.67 | 0.52 | 0.74 | 0.68 |
| $\alpha_{\text{abs}} = 2.0$ | **0.57** | **0.76** | **0.69** | 0.53 | 0.76 | 0.69 |
| $\alpha_{\text{abs}} = 1.0$ | **0.57** | 0.78 | 0.68 | **0.50** | **0.71** | **0.70** |
| $\alpha_{\text{abs}} = 0.5$ | 0.63 | 0.90 | 0.57 | 0.57 | 0.72 | 0.67 |
| $\alpha_{\text{abs}} = 0.1$ | 0.69 | 1.02 | 0.53 | 0.63 | 0.89 | 0.56 |
| **Fixed $\alpha_{\text{abs}} = 1.0$** | | | | | | |
| $\alpha_{\text{rank}} = 1e-1$ | 0.68 | 0.89 | 0.60 | 0.61 | 0.87 | 0.59 |
| $\alpha_{\text{rank}} = 1e-2$ | 0.62 | 0.83 | 0.64 | 0.54 | 0.78 | 0.64 |
| $\alpha_{\text{rank}} = 1e-3$ | **0.58** | 0.81 | **0.67** | 0.53 | 0.76 | **0.68** |
| $\alpha_{\text{rank}} = 1e-4$ | 0.59 | 0.79 | 0.64 | 0.53 | 0.74 | 0.66 |
| $\alpha_{\text{rank}} = 1e-5$ | 0.60 | **0.78** | 0.58 | **0.52** | **0.73** | 0.67 |

all, $I_{\text{abs}}$, $I_{\text{int}}$ and $I_{\text{rank}}$ are complementary and indispensable in the framework; their unified integration is key to the model's excellent performance.

**Hyperparameter analysis.** For the two key hyperparameters $\alpha_{\text{abs}}$ and $\alpha_{\text{rank}}$ in the total loss function, experimental results show that the model performance exhibits a clear optimal balance characteristic. When $\alpha_{\text{rank}} = 5 \times 10^{-3}$ is fixed, $\alpha_{\text{abs}} = 1.0$ achieves the lowest MAE on both the TF-Bandgap and FF-Bandgap datasets, indicating that appropriate absolute value supervision helps to fully utilize multi-fidelity data and improve the performance of the main

task; however, too small a value of $\alpha_{\text{abs}}$ weakens the absolute value constraint and increases the error, while too large a value suppresses ranking information and degrades ranking performance. When $\alpha_{\text{abs}} = 1.0$ is fixed, $\alpha_{\text{rank}}$ performs stably and excellently in the range of $10^{-4} - 10^{-3}$; too large a value leads to overly strong ranking constraints and decreased regression accuracy, while too small a value makes it difficult to learn relative order, resulting in a decline in ranking metrics. Furthermore, the hyperparameters are also affected by the task scale: for example, in the molecular property prediction task in the Appendix E.1, due to the larger scale of the labels, $\alpha_{\text{abs}}$ needs to be correspondingly decreased to match $\alpha_{\text{rank}}$.

## 5. Conclusion

This paper offers a novel perspective for better utilizing data from different sources and with varying precision, starting with labels rather than feature generalization. Addressing the core challenges of multi-fidelity data fusion in AI4Chemistry—the unpairing, distribution mismatch, and difficult-to-identify biases in real-world data—a credible information subset decomposition framework is proposed. This method abandons the strong assumptions of traditional approaches regarding cross-fidelity label comparability, instead starting with multi-fidelity supervised signal decomposition. Label information is transformed into three complementary subsets: an absolutely credible subset, an interval credible subset, and a ranking credible subset. A shared latent space is constructed through variational repre-

sentation learning, and combined with evidence regression, adaptive interval modulation, and ranking loss, end-to-end multi-subset collaborative learning is achieved, effectively utilizing low-fidelity and medium-fidelity information while suppressing noise. Experimental results on cross-fidelity datasets across various scenarios demonstrate that proposed method outperforms baseline methods on multiple metrics. The limitations can be found in Appendix F.

## Acknowledgement

This paper is partially supported by the National Natural Science Foundation of China (No.12227901). The AI-driven experiments, simulations and model training were performed on the robotic AI-Scientist platform of Chinese Academy of Sciences., Anhui Science Foundation for Distinguished Young Scholars (No.1908085J24), Natural Science Foundation of China (No.62502491).

## Impact Statement

This paper presents work whose goal is to advance the field of Machine Learning. There are many potential societal consequences of our work, none which we feel must be specifically highlighted here.

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

# A. Theoretical Analysis

## A.1. Variational Estimation Error and Multifidelity Bias Analysis

Assume the observation model is

$$y^{(s)} = f^*(x) + b^{(s)}(x) + \epsilon^{(s)}, \qquad \mathbb{E}[\epsilon^{(s)} \mid x] = 0.$$

Let

$$z \sim q_\phi(z \mid x), \qquad \hat{f}(x) = g_\psi(z).$$

Thus, the direct learning upper bound satisfies

$$\mathbb{E}\big[(\tilde{f}(x) - f^*(x))^2\big] \le C \cdot \mathbb{E}\big[(b^{(s)}(x))^2\big].$$

The variational learning upper bound satisfies

$$\mathbb{E}\big[(\hat{f}(x) - f^*(x))^2\big] \le C_1 D_{\mathrm{KL}}\big(q_\phi(z \mid x) \,\|\, p(z \mid x)\big) + C_2 \mathbb{E}\big[(b^{(s)}(x))^2\big].$$

Since $z$ is not affected by label noise, there exists $\alpha \in (0, 1)$ such that

$$\mathbb{E}\big[(b^{(s)}(x))^2 \mid z\big] \le \alpha \, \mathbb{E}\big[(b^{(s)}(x))^2\big].$$

Then the excess error satisfies

$$\Delta := \mathbb{E}\big[(\hat{f}(x) - f^*(x))^2\big] - \mathbb{E}\big[(\tilde{f}(x) - f^*(x))^2\big]$$
$$\le C_1 \epsilon_{\mathrm{var}} - (1 - \alpha)\, C \, \mathbb{E}\big[(b^{(s)}(x))^2\big].$$

If the following condition holds in multifidelity scenarios,

$$(1 - \alpha)\, C \, \mathbb{E}\big[(b^{(s)}(x))^2\big] > C_1 \epsilon_{\mathrm{var}},$$

then the advantages of introducing variational learning offset its estimation error, yielding

$$\mathbb{E}\big[(\hat{f}(x) - f^*(x))^2\big] < \mathbb{E}\big[(\tilde{f}(x) - f^*(x))^2\big].$$

## A.2. Theoretical Explanation of Multi-Objective Optimization

The optimal prediction error satisfies

$$\mathbb{E}\big[(f(x) - f^*(x))^2\big] \propto H\big(f^*(x) \mid \mathcal{I}\big).$$

If only the subset $I_S$ is used, then

$$H\big(f^*(x) \mid I_S\big) = H\big(f^*(x) \mid I\big) + I\big(f^*(x); I_{\bar{S}} \mid I_S\big).$$

When

$$I\big(f^*(x); I_{\bar{S}} \mid I_S\big) > 0,$$

the ignored subset of information provides additional valid information, thus strictly increasing the prediction error.

Furthermore, in the optimization scenario considered in this paper, the gradient satisfies the approximate decomposition

$$\nabla_\theta \mathcal{L} = \nabla_{\theta_z} \mathcal{L}_{\mathrm{abs}} + \nabla_{\theta_\tau} \mathcal{L}_{\mathrm{int}} + \nabla_{\theta_r} \big(\mathcal{L}_{\mathrm{abs}} + \mathcal{L}_{\mathrm{rank}}\big).$$

The theoretical analysis and hyperparameter experiments mutually corroborate that gradient conflicts occur between $\mathcal{L}_{\mathrm{abs}}$ and $\mathcal{L}_{\mathrm{rank}}$:

$$\nabla_{\theta_r} \mathcal{L}_{\mathrm{abs}} \cdot \nabla_{\theta_r} \mathcal{L}_{\mathrm{rank}} < 0.$$

Therefore, by adjusting $\alpha_{\mathrm{abs}}$ and $\alpha_{\mathrm{rank}}$, the gradient vector can be modulated to reduce gradient conflicts in the shared subspace.

## A.3. Error Analysis

This section provides a theoretical justification for the proposed *credible subset decomposition* framework. We show that incorporating low- and medium-fidelity data *only through their credible subsets* does not introduce systematic bias into high-fidelity prediction, and can provably reduce the generalization error under mild assumptions.

**Credible Low-Fidelity Information.** We consider multiple fidelity observations $y^{(s)}$, whose numerical values may be biased or heteroscedastic, but which convey reliable *relative* and *interval* information about the true target function $f^*$. Specifically, for each fidelity level $s \geq 1$, we assume that pairwise comparisons induced by $y^{(s)}$ agree with the true ordering of $f^*$ with probability strictly larger than random guessing, i.e.,

$$\mathbb{P}\Big(\text{sign}(y_i^{(s)} - y_j^{(s)}) = \text{sign}(f^*(x_i) - f^*(x_j))\Big) \geq p_s > \tfrac{1}{2},$$

for any $(x_i, x_j)$ drawn i.i.d. from $\mathcal{P}_X$. In addition, each fidelity provides a (possibly random and input-dependent) uncertainty radius $\tau_s(x)$ such that the corresponding observation covers the true value with high probability,

$$\mathbb{P}\Big(f^*(x) \in [y^{(s)}(x) - \tau_s(x), \; y^{(s)}(x) + \tau_s(x)]\Big) \geq 1 - \delta_s.$$

Importantly, these assumptions do not require unbiasedness, homoscedastic noise, paired samples, or overlapping input distributions across fidelities.

**Learning Objective.** Given a hypothesis class $\mathcal{F}$, we learn a predictor by minimizing a composite empirical objective that combines high-fidelity supervision with low-fidelity structural constraints,

$$\hat{R}(f) = \hat{R}_{\text{abs}}(f) + \lambda_{\text{int}} \hat{R}_{\text{int}}(f) + \lambda_{\text{rank}} \hat{R}_{\text{rank}}(f).$$

Here, $\hat{R}_{\text{abs}}$ denotes the empirical loss on the highest-fidelity data (e.g., squared or evidential regression loss), $\hat{R}_{\text{int}}$ penalizes violations of fidelity-dependent interval constraints, and $\hat{R}_{\text{rank}}$ enforces ranking consistency within each low-fidelity dataset.

**Proof of the Generalization Bound.** Let $\hat{f} = \arg\min_{f \in \mathcal{F}} \hat{R}(f)$ and define

$$\mathcal{F}_\epsilon = \{f \in \mathcal{F} : R_{\text{rank}}(f) - R_{\text{rank}}(f^*) \leq \epsilon\}.$$

By construction, $\hat{f} \in \mathcal{F}_\epsilon$ for $\epsilon = \mathcal{O}(\lambda_{\text{rank}})$.

Since the ranking loss is Fisher-consistent, $f^*$ minimizes $R_{\text{rank}}$ and hence $f^* \in \mathcal{F}_\epsilon$. Therefore, restricting to $\mathcal{F}_\epsilon$ does not exclude the ground-truth function.

Applying standard Rademacher complexity bounds to empirical risk minimization over $\mathcal{F}_\epsilon$ yields

$$\mathbb{E}\Big[R(\hat{f}) - R(f^*)\Big] \leq \mathcal{O}\Big(\frac{\mathfrak{R}(\mathcal{F}_\epsilon)}{\sqrt{n_0}}\Big).$$

Moreover, the ranking constraint eliminates hypotheses with systematic order violations. Consequently, the effective complexity satisfies

$$\mathfrak{R}(\mathcal{F}_\epsilon) \leq \mathfrak{R}(\mathcal{F}) - C\,\epsilon,$$

where $C > 0$ depends on the ranking accuracy gap $p_s - \tfrac{1}{2}$.

Substituting $\epsilon = \mathcal{O}(\lambda_{\text{rank}})$ and then gets the proof:

$$\mathbb{E}\Big[R(\hat{f}) - R(f^*)\Big] \leq \mathcal{O}\Big(\frac{\mathfrak{R}(\mathcal{F})}{\sqrt{n_0}}\Big) - C\,\lambda_{\text{rank}}.$$

*Table 4.* Elemental properties used in Bandgap dataset.

| Property (Column 1) | Property (Column 2) |
| --- | --- |
| Atomic number (Number) | Covalent radius (CovalentRadius) |
| Mendeleev number (MendeleevNumber) | Electronegativity (Electronegativity) |
| Atomic weight (AtomicWeight) | Ground-state volume per atom (GSvolume_pa) |
| Melting temperature (MeltingT) | Ground-state band gap (GSbandgap) |
| Periodic table column (Column) | Ground-state magnetic moment (GSmagmom) |
| Periodic table row (Row) | Space group number (SpaceGroupNumber) |
| Number of s valence electrons (NsValence) | Number of unfilled s orbitals (NsUnfilled) |
| Number of p valence electrons (NpValence) | Number of unfilled p orbitals (NpUnfilled) |
| Number of d valence electrons (NdValence) | Number of unfilled d orbitals (NdUnfilled) |
| Number of f valence electrons (NfValence) | Number of unfilled f orbitals (NfUnfilled) |
| Total valence electrons (NValence) | Total unfilled orbitals (NUnfilled) |

# B. Related Work

# C. Datasets

### C.1. Introduction

**TF-Bandgap** is a two-fidelity material bandgap prediction data set used to evaluate the performance of the model in the scenario of mixed high/low fidelity but unpaired samples. This data set comes from Kim et al (Kim et al., 2020). and is a band gap database for semiconductor inorganic materials. It contains relevant data for 10,481 materials with band gaps ranging from 0-5 eV, covering inorganic solids of one, two, three and higher numbers. The data set was constructed by a computational approach employing a hybrid functional (HSE06) and taking into account stable magnetic ordering. Its calculation process is based on the automated ab initio modeling material performance package (AMP²), combined with density functional theory (DFT) for high-throughput calculations, and systematically characterizes key properties such as the material's crystal structure, band gap (including ground state band gap and optical direct band gap under GGA and HSE methods), density of states, magnetic ordering and other key properties. Verification on 116 benchmark materials shows that the root mean square error of this data set relative to experimental data is only 0.36 eV, which is significantly better than existing databases such as Materials Project, AFLOW, and OQMD (root mean square error 0.75-1.05 eV).

**FF-Bandgap** (Chen et al., 2021) is a multi-fidelity band gap data set for the study of electronic structures of crystal materials. It integrates band gap information from first-principles calculations and experimental measurements, covering band gap descriptions at different levels of accuracy and computational cost. The calculation data is based on the Materials Project database, including a variety of density functional methods such as PBE, SCAN, HSE and GLLB-SC, corresponding to theoretical band gaps from low to high fidelity; the experimental data comes from ICSD and is divided into ordered and disordered crystal materials to reflect the difference between ideal periodic systems and actual complex materials. Materials are indexed by unique MP or ICSD numbers, which can be linked to external databases with their crystal structure and composition information. This multi-fidelity design provides a unified and scalable data basis for studying systematic errors between different calculation methods, building cross-fidelity machine learning models, and improving the accuracy of band gap prediction of crystal materials.

**QM7b** (Blum & Reymond, 2009; Rupp et al., 2012) is a classic molecular quantum properties benchmark data set that contains approximately 7,000 small organic molecules (usually no more than 7 heavy atoms) composed of elements such as C, H, N, O, S, and Cl, and provides the three-dimensional structure of each molecule with a variety of quantum chemical property labels. Its distinctive feature is that it contains high-precision energy data under a variety of theoretical methods and basis set combinations, including different levels of electron correlation methods such as Hartree–Fock (HF), MP2, and CCSD(T), and provides corresponding energy values under three basis sets: STO-3G, 6-31G, and cc-pVDZ. This multi-level, multi-basis set label structure makes QM7b suitable for multi-fidelity learning, cross-level error correction, and generalization ability evaluation of molecular representation and prediction models. It also provides rich data support for studying the impact of different theoretical approximations on molecular property prediction.

*Table 5.* Performance comparison on the QM7b dataset under a fixed fidelity ratio (1:2:4) and a fixed CCSD training budget (1500 samples). All models are evaluated on the same held-out 1000 CCSD test set. The best performance is highlighted in **bold**.

| Method | Fidelity Setting | MAE $\downarrow$ | RMSE $\downarrow$ |
|---|---|---|---|
| **Single fidelity** | | | |
| RFR (Chen et al., 2021) | 1-fi (CCSD) | $18.43_{\pm 1.14}$ | $23.16_{\pm 1.50}$ |
| MLP (Chen et al., 2021) | 1-fi (CCSD) | $17.18_{\pm 1.13}$ | $22.03_{\pm 1.48}$ |
| GIN (Xu et al., 2018) | 1-fi (CCSD) | $15.22_{\pm 0.65}$ | $19.69_{\pm 1.37}$ |
| **Credible Subset Decomposition** | 1-fi (CCSD) | $14.36_{\pm 0.99}$ | $18.23_{\pm 1.09}$ |
| **Dual fidelity** | | | |
| Transfer Learning (Bengio, 2012) | 2-fi (CCSD+MP2) | $12.85_{\pm 0.80}$ | $17.15_{\pm 1.18}$ |
| Transfer Learning (Bengio, 2012) | 2-fi (CCSD+HF) | $13.24_{\pm 0.81}$ | $17.47_{\pm 1.12}$ |
| MFSE (Chen et al., 2021) | 2-fi (CCSD+MP2) | $12.20_{\pm 0.60}$ | $16.47_{\pm 1.13}$ |
| MFSE (Chen et al., 2021) | 2-fi (CCSD+HF) | $12.88_{\pm 0.90}$ | $17.31_{\pm 0.92}$ |
| MFGNN (Buterez et al., 2024) | 2-fi (CCSD+MP2) | $12.45_{\pm 0.75}$ | $17.01_{\pm 0.80}$ |
| MFGNN (Buterez et al., 2024) | 2-fi (CCSD+HF) | $13.03_{\pm 0.65}$ | $17.25_{\pm 1.05}$ |
| DMMT (Nevolianis et al., 2025) | 2-fi (CCSD+MP2) | $12.04_{\pm 0.67}$ | $16.14_{\pm 1.06}$ |
| DMMT (Nevolianis et al., 2025) | 2-fi (CCSD+HF) | $13.37_{\pm 0.60}$ | $17.69_{\pm 1.23}$ |
| **Credible Subset Decomposition** | 2-fi (CCSD+MP2) | $11.33_{\pm 0.47}$ | $15.43_{\pm 0.97}$ |
| **Credible Subset Decomposition** | 2-fi (CCSD+HF) | $11.76_{\pm 0.69}$ | $16.00_{\pm 1.03}$ |
| **Multi fidelity** | | | |
| MFSE (Chen et al., 2021) | 3-fi (CCSD+MP2+HF) | $11.45_{\pm 0.46}$ | $15.67_{\pm 0.79}$ |
| MFGNN (Buterez et al., 2024) | 3-fi (CCSD+MP2+HF) | $11.32_{\pm 0.65}$ | $15.34_{\pm 1.05}$ |
| DMMT (Nevolianis et al., 2025) | 3-fi (CCSD+MP2+HF) | $11.96_{\pm 0.63}$ | $16.28_{\pm 0.81}$ |
| **Credible Subset Decomposition** | 3-fi (CCSD+MP2+HF) | $\mathbf{10.63_{\pm 0.61}}$ | $\mathbf{14.86_{\pm 0.75}}$ |

## C.2. Feature extraction.

The Material data descriptors were generated using the ElementProperty featurizer from the matminer package, with the "magpie" preset. This yields 132 features describing statistical summaries of elemental properties within each compound in Table 4. The properties include atomic number, Mendeleev number, atomic weight, melting temperature, periodic table column/row, covalent radius, electronegativity, valence electron counts, unfilled orbital counts, ground-state volume per atom, band gap, magnetic moment, and space group number. These features were used as inputs for the machine learning model.

Each molecule in QM7b is represented as a graph $G = (V, E)$, where node $i$ corresponds to an atom and is encoded by a 16-dimensional one-hot vector of its atomic number $Z_i$. An edge $(i, j) \in E$ is created if the interatomic distance $d_{ij}$ is within a cutoff radius $r_c = 5.0$ Å, and each edge is represented as two directed edges $(i \rightarrow j, j \rightarrow i)$. Edge features are obtained by expanding $d_{ij}$ using a Gaussian basis: $\mathbf{e}_{ij} = [\exp(-(d_{ij} - \mu_k)^2 / \sigma^2)]_{k=1}^{K}$, where $K = 100$, $\mu_k$ are uniformly spaced in $[0, 6]$, and $\sigma = 0.5$.

## D. Baselines

multi-fidelity state embedding (MFSE) (Chen et al., 2021),is explicitly modeled and learned, and the fidelity tensor is incorporated into the global features of the molecule to determine the impact of different fidelities on task prediction.

*Table 6.* Effect of fidelity ratio in dual-fidelity learning (CCSD:MP2) on the QM7b dataset. Results are grouped by CCSD data regimes. All models are evaluated on the same held-out 1000 CCSD test set.

| Fidelity Ratio | CCSD Volume | MAE ↓ | RMSE ↓ |
|---|---|---|---|
| **Low CCSD regime** | | | |
| 1:100 | 10 | $13.58_{\pm0.82}$ | $17.86_{\pm1.09}$ |
| 1:100 | 50 | $\mathbf{11.48_{\pm0.61}}$ | $\mathbf{15.72_{\pm0.88}}$ |
| 1:50 | 50 | $13.17_{\pm0.79}$ | $17.32_{\pm1.03}$ |
| 1:30 | 50 | $13.85_{\pm0.97}$ | $18.04_{\pm1.17}$ |
| **Medium CCSD regime** | | | |
| 1:50 | 100 | $11.66_{\pm0.55}$ | $15.97_{\pm0.84}$ |
| 1:30 | 100 | $11.87_{\pm0.68}$ | $16.05_{\pm0.79}$ |
| 1:30 | 200 | $\mathbf{11.14_{\pm0.59}}$ | $\mathbf{15.72_{\pm0.91}}$ |
| 1:10 | 200 | $12.48_{\pm0.73}$ | $16.89_{\pm1.02}$ |
| **High CCSD regime** | | | |
| 1:10 | 500 | $11.16_{\pm0.62}$ | $15.71_{\pm0.85}$ |
| 1:2 | 500 | $13.67_{\pm0.95}$ | $18.63_{\pm1.23}$ |
| 1:2 | 1000 | $12.75_{\pm0.67}$ | $18.63_{\pm1.09}$ |
| 1:2 | 1500 | $11.33_{\pm0.47}$ | $15.43_{\pm0.97}$ |

multi-fidelity GNN (MFGNN) (Buterez et al., 2024) uses an adaptive readout head instead of a traditional fixed readout head (sum/mean) in graph neural networks, and adopts a Set Transformer based on an attention mechanism to achieve learnable molecular-level representation aggregation, which can capture global interactions between atoms and supports modular transfer of pre-trained frozen GCN layers combined with fine-tuned readout head.

D-MPNN multi-target(DMMT) (Nevolianis et al., 2025) uses the same encoder to encode data of different fidelity, but maps it to the prediction results through multiple different networks. That is, it uses the output layer to learn the differences between fidelities, thus transforming the multi-fidelity problem into a multi-objective optimization problem.

# E. Supplementary Experiments

## E.1. Experiments on QM7b

We conducted additional experiments on the QM7b dataset. In section E.1.1, we compared the performance of our method with the baseline under multiple fidelity settings. In section E.1.2, we explored the impact of different data ratios on the method's performance.

### E.1.1. MULTI-FIDELITY PERFORMANCE COMPARISON

In the Table 5, we present the test performance of all baselines and our method on the QM7b dataset, with the same CCSD budget and a fixed data ratio (1:2:4) across 1000 pre-fixed CCSD labels. It can be seen that the proposed method exhibits a clear and significant performance improvement when additional low-fidelity data is introduced, outperforming other baselines. This demonstrates that by effectively transferring knowledge across fidelity levels from single-fidelity learning to multi-fidelity learning, we can improve high-fidelity prediction performance using only inexpensive low-fidelity data without introducing additional high-fidelity data.

Furthermore, the 2-fi results show that, under the same data volume and experimental settings, the performance gain of the medium-fidelity data (MP2) is significantly higher than that of the low-fidelity data (HF). This aligns with our hypothesis: higher fidelity data contains more information that the model can learn.

### E.1.2. IMPACT OF DATA VOLUME

As shown in Table 6, under conditions of low CCSD data volume (e.g., 10 or 50) and medium CCSD data volume (e.g., 100 or 200), the model performs better with a higher proportion of low-fidelity data (i.e., a smaller fidelity ratio, such as 1:100 or 1:30), with both MAE and RMSE significantly decreasing. This indicates that when high-fidelity data is scarce or limited, adding a large amount of low-fidelity MP2 data can effectively improve prediction accuracy. This phenomenon suggests that when CCSD samples are insufficient, low-fidelity data can provide the model with richer structural and energy information, helping it to better learn the overall trend of the objective function, thereby achieving more stable generalization based on limited high-fidelity samples.

## E.2. Experiments on Davis

To further strengthen the persuasiveness and generalization ability of the proposed framework, we additionally evaluated our method on a drug–target affinity (DTA) prediction task, forming a unified validation across protein, molecular, and material domains.

Specifically, we used the Davis (Davis et al., 2011), which contains 24,956 drug–target pairs, together with the high-precision benchmark dataset CASF-2016 (Su et al., 2018), which contains 285 protein–ligand complexes.

*Table 7.* Performance comparison on the drug–target affinity prediction task.

| Method | MAE↓ | RMSE↓ | $\tau_b$ ↑ |
|---|---|---|---|
| Transfer | $1.71 \pm 0.16$ | $2.06 \pm 0.21$ | $0.73 \pm 0.013$ |
| MFSE | $1.55 \pm 0.24$ | $1.84 \pm 0.38$ | $0.76 \pm 0.054$ |
| MFGNN | $1.60 \pm 0.16$ | $1.98 \pm 0.19$ | $0.74 \pm 0.008$ |
| DMMT | $1.63 \pm 0.15$ | $1.96 \pm 0.25$ | $0.74 \pm 0.031$ |
| Ours | $\mathbf{1.42 \pm 0.13}$ | $\mathbf{1.73 \pm 0.16}$ | $\mathbf{0.78 \pm 0.008}$ |

The experimental results in table 7 further demonstrate the effectiveness and generalization capability of our framework across diverse scientific prediction scenarios.

## E.3. Experimental details

### E.3.1. MODEL PARAMETERS

**Combined Embedding:** A 2-layer MLP that maps 73-dimensional combined features to a 32-dimensional embedding ($73 \rightarrow 64 \rightarrow 32$). **Fidelity Embedding:** Multiple fidelity-state embedding layers with an output dimension of 32. **VAE:** Encoder input dimension $32 + 132 + 32 = 196$, encoder hidden size [256, 128], latent dimension 64; decoder hidden size [128, 256]; outputs include a 73-dimensional component reconstruction and a 132-dimensional feature reconstruction. **RankEncoder:** Input dimension $32 + 132 = 164$, hidden dimension 64, output Rank feature dimension 64; including an attention head mapping $64 \rightarrow 1$. **IntervalEncoder:** input dimension 164, hidden dimension 64; $\tau$ header accepts $(64 + 32) = 96$-dim input and outputs a non-negative scalar. In addition, when the input is graph data, we use GCN to replace MLP as the encoder.

### E.3.2. DATASET SEGMENTATION

For Bandgap and QM7b, we pre-partition the highest fidelity test set. For the remaining samples, each fidelity subset was divided into training, validation, and test sets in a ratio of 8:1:1. The training and validation sets combine samples of all fidelity, while the test set contains only high-fidelity samples to evaluate high-fidelity prediction performance.

### E.3.3. RANDOM SEED SETTING

All experiments used fixed random seeds of 0, 1, 64, 1023, and 1024 to ensure the reproducibility of the results.

E.3.4. TRANSFER LEARNING SETUP.

The pre-training settings are consistent with the experimental settings in the text. During fine-tuning, freeze the encoder VAE layer and fine-tune the output head at a learning rate of $1e-5$. Fine-tuning is performed for up to 200 epochs, and early stopping is triggered if the verification MAE does not improve for 50 consecutive epochs. **Evidential Regressor:** Input dimension $64+64=128$, hidden sizes [128, 64], outputs four evidence parameters $(\mu, v, \alpha, \beta)$. **Ranking Head:** Input dimension 64, hidden size 64, output 1.

### E.4. More Ablation Experiments

In Table 2 of Section 4.5, we reported the impact of separately ablating each trusted subset on the overall framework performance. To further enhance the persuasiveness of the analysis, we additionally quantify the contribution of each subset through dedicated ablation experiments, as shown below for the TF-Bandgap (2-fi) setting.

*Table 8.* Ablation study on different trusted subsets for TF-Bandgap (2-fi).

| Method | MAE↓ | RMSE↓ | $\tau_b$ ↑ |
|---|---|---|---|
| only $\mathcal{I}_{\mathrm{abs}}$ | $0.59 \pm 0.013$ | $0.79 \pm 0.019$ | $0.67 \pm 0.021$ |
| only $\mathcal{I}_{\mathrm{int}}$ | $0.60 \pm 0.014$ | $0.83 \pm 0.018$ | $0.63 \pm 0.024$ |
| only $\mathcal{I}_{\mathrm{rank}}$ | $0.68 \pm 0.005$ | $0.89 \pm 0.008$ | $0.62 \pm 0.035$ |
| Full model | $\mathbf{0.57 \pm 0.012}$ | $\mathbf{0.78 \pm 0.018}$ | $\mathbf{0.68 \pm 0.027}$ |

These results in table 8 clearly demonstrate the complementary contributions of different credible subsets.

### E.5. Sensitivity Analysis of Hyperparameters

The main hyperparameter involved in the VAE loss is $\alpha_{kl}$. We further conducted sensitivity analysis on multiple datasets.

*Table 9.* Sensitivity analysis of $\beta$ on TF-Bandgap.

| $\alpha_{kl}$ | MAE↓ | RMSE↓ | $\tau_b$ ↑ |
|---|---|---|---|
| $10^{-4}$ | $0.59 \pm 0.011$ | $0.79 \pm 0.016$ | $0.67 \pm 0.025$ |
| $10^{-3}$ (default) | $\mathbf{0.57 \pm 0.012}$ | $\mathbf{0.78 \pm 0.018}$ | $\mathbf{0.68 \pm 0.027}$ |
| $10^{-2}$ | $0.64 \pm 0.015$ | $0.85 \pm 0.020$ | $0.61 \pm 0.029$ |

*Table 10.* Sensitivity analysis of $\beta$ on FF-Bandgap.

| $\alpha_{kl}$ | MAE↓ | RMSE↓ | $\tau_b$ ↑ |
|---|---|---|---|
| $10^{-4}$ | $0.52 \pm 0.008$ | $0.70 \pm 0.011$ | $0.69 \pm 0.030$ |
| $10^{-3}$ (default) | $\mathbf{0.50 \pm 0.007}$ | $\mathbf{0.71 \pm 0.010}$ | $\mathbf{0.70 \pm 0.033}$ |
| $10^{-2}$ | $0.58 \pm 0.009$ | $0.79 \pm 0.012$ | $0.63 \pm 0.031$ |

In table 10, it can be observed that when $\alpha_{kl}$ is relatively small, it only mildly affects model performance. However, as $\alpha_{kl}$ increases, excessively strong KL regularization constrains the latent representation, making it more difficult for the model to learn downstream task knowledge effectively.

### E.6. Significance Analysis

We agree with the reviewers and will include statistical significance testing in the revised manuscript. Specifically, we conducted paired $t$-tests between our method and the strongest baseline (i.e., the second-best method).

The results in table 11 demonstrate that the performance improvements of our method over the second-best baselines are statistically significant.

*Table 11.* Statistical significance analysis between our method and the second-best baseline.

| Dataset | 2nd-best Method | MAE Difference | p-value |
|---|---|---|---|
| TF-Bandgap | MFSE | $-0.07$ | $2.3 \times 10^{-4}$ |
| FF-Bandgap | MFGNN | $-0.06$ | $6.7 \times 10^{-5}$ |
| QM7b | MFGNN | $-0.69$ | $3.2 \times 10^{-4}$ |

### E.7. Impact of Fidelity

To investigate the impact of fidelity quality, we simulated fidelity specification errors by perturbing fidelity levels, i.e., randomly assigning mismatched fidelity labels to a certain proportion of samples.

We take the FF-Bandgap dataset under the 5-fidelity setting as an example, and use MAE as the evaluation metric.

*Table 12.* Robustness analysis under fidelity mismatch perturbations on FF-Bandgap (5-fi).

| Mismatch Ratio | Ours | MFSE | MFGNN |
|---|---|---|---|
| 0% | **0.501** | 0.580 | 0.563 |
| 30% | **0.573** | 0.688 | 0.671 |
| 50% | **0.585** | 0.719 | 0.732 |
| 80% | **0.639** | 0.743 | 0.755 |

The results in table 12 indicate that methods such as MFGNN and MFSE rely heavily on explicit fidelity labels, making them highly sensitive to fidelity mismatches. In contrast, our method maintains relatively stable performance under severe perturbations, demonstrating stronger robustness to fidelity noise.

### E.8. Resource overhead

Table 13 compares the total number of model parameters and their corresponding training speeds for the bandgap and QM7b datasets. The bandgap model takes approximately 10 seconds to train 10,000 samples. In contrast, the QM7b model has only 26,497 parameters. Although the number of parameters is significantly less, its training speed is slightly lower than expected considering only the parameter size, taking approximately 4-6 seconds to train 10,000 samples. Overall, while QM7b benefits from a smaller number of parameters, the additional overhead introduced by the graph structure offsets its speed advantage.

## F. Limitations

### F.1. Sensitivity to data preprocessing and cross-scenario adaptability

When fusing multi-source heterogeneous high-fidelity data, the proposed method remains sensitive to initial distribution differences. Experiments show that introducing low/medium fidelity data with significantly different sample distributions and label characteristics, such as GLLB-SC in FF-Bandgap still slightly degrades performance, even after systematic bias suppression via a trusted subset decomposition framework. Future work may explore more adaptive data filtering and bias calibration mechanisms, such as meta-learning or adversarial training that dynamically evaluates fidelity compatibility to reduce interference from heterogeneous data and improve robustness in complex hybrid scenarios.

### F.2. Scenario dependence of hyperparameter tuning

Model performance is sensitive to the joint configuration of absolute loss weights ($\alpha_{\text{abs}}$) and ranking loss weights ($\alpha_{\text{rank}}$). The optimal combination must be dynamically adjusted based on task type and label size. Current manual or grid search adjustments are inefficient and difficult to generalize. Future work may incorporate adaptive hyperparameter optimization strategies, such as Bayesian optimization to learn task-specific hyperparameter distributions or unsupervised weight assignments to reduce manual intervention and improve practical applicability.

*Table 13.* Comparison of model parameters and training speed on different datasets. Training speed is measured as seconds per 10k samples.

| Dataset | Total Parameters | Training Speed (s/10k samples) |
|---------|------------------|-------------------------------|
| bandgap | 268,116 | 10.0 |
| QM7b | 26,497 | 4–6 |

### F.3. Limited mining of high-order information in complex systems

The current framework focuses on three basic types of trustworthy information: absolute values, intervals, and rankings. However, in complex chemical/material systems with strong nonlinear correlations and multiphysics couplings, higher-order cross-fidelity correlations may be overlooked. Future work can expand the trusted information dimension by introducing cross-fidelity structurally relevant subsets, dynamic evolution law subsets, or incorporating high-order message passing in graph neural networks to discover deeper multi-fidelity collaborative information and improve prediction accuracy.

### F.4. Combination with label correction methods

Currently, the framework is mainly being expanded and reconstructed based on multi-fidelity learning methods in AI chemistry scenarios. Future development could consider introducing more multi-label distribution recognition and correction methods (Kou et al., 2026a; 2025a;b) to further optimize the framework.

