# OpenReview forum: "Credible Information Subset Decomposition: An End-to-End Multi-fidelity Learning Model by Modeling Label Information"
_ICML.cc/2026/Conference — ICML 2026 regular_

### Official Review · Reviewer_4EAA · 2026-03-04

**Soundness:** 3
**Presentation:** 2
**Significance:** 2
**Originality:** 3
**Overall Recommendation:** 3
**Confidence:** 4

**Summary:**

This paper considers multi fidelity dataset in AI4Chemistry scenario. As high fidelity data are reliable and scarce while low fidelity data are unreliable and abundant, this paper tries to use the benefits from both of two data. As previous methods require paired samples between high fidelity and low fidelity data which is not realistic, this paper suggests a new framework to utilize labels with various fidelity in hierarchical structure. Experiments show good performances.

**Compliance With Llm Reviewing Policy:**

Affirmed.

**Final Justification:**

I have read the rebuttal, but I am maintaining my original score. While the empirical results are supportive, the core theoretical concerns remain unaddressed:

1. The authors rely on empirical intuition rather than providing a principled explanation for using a VAE-based latent space despite its intrinsic estimation errors.

2. The specific combination of three loss functions lacks an optimization-theoretic justification and an analysis of potential gradient interference.

3.The assumption that fidelity is "clear from the source" is a human bias, not a methodological solution to handling data noise.

Overall, the paper presents a collection of empirical tricks without the rigorous theoretical foundation.

**Key Questions For Authors:**

-Why did the authors construct a shared latent space through variational representation learning? I mean, as VAE is a variational inference learning framework to learn the distribution of latent variable which is unknown, it should contain estimation error. Why the representation should be learned as latent?
-How can we trust the given fidelity of each dataset? I think the confidence of fidelity is very essential for subset decomposition. If the real fidelity of each data is differernt from the given fidelity, the subset learning can be miselad.

**Limitations:**

See above.

**Strengths And Weaknesses:**

- The method suggested in this paper does not requrie the paired samples between high and low fidelity, which may be practical.
- Legends of figures are too small.
- Applying the mixture of several tricks to solve the diverse fidelity problem can be novel, but each loss function is already given.
- Why each loss should be chosen is not fully explained. It is not clear why each modeling are required.
- If combinations of losses are used, each loss may disturb other losses. Analyses of what each loss term is doing are required (including ablation studies, sensitivity analysis).

---

> ### Author Rebuttal · Authors · 2026-03-29
>
> Thanks for your thoughtful and constructive feedback. We sincerely appreciate the time and effort you invested in reviewing our manuscript. Below we provide a detailed, point-by-point response to your comments and suggestions.
> ___
>
> > **W1.** Figure readability
>
> We sincerely apologize for any inconvenience caused by our drawing error. We have carefully reviewed the drawing issues. **In the revised manuscript, we promise to convert all drawings to vector images and enlarge the legends and coordinate information.
>
> > **W2.** Loss Function Design and Model Selection
>
> Our core innovation lies in the credibility-aware decomposition of multi-fidelity labels, rather than a single loss term. We unify evidence regression, interval modulation, and ranking loss under a hierarchical credibility subset paradigm. **Each loss is targeted at a different type of credible information, thus forming a complementary framework, rather than a simple combination of existing techniques.**
>
> We provide clear theoretical and empirical motivations:
>
> - **Absolute loss: Models uncertainty of high-fidelity labels, avoiding uniform MSE bias.
> - **Interval loss**: Characterizes error ranges of medium-fidelity data, relaxing rigid point constraints.
> - **Rank loss**: Captures stable order relations preserved in low-fidelity data, robust to scale distortion.
> - **VAE**: Learns fidelity-agnostic input structure, preventing low-fidelity noise from leaking into representations.
>
> > **W3.** Ablation&Sensitivity Analysis
>
> We have conducted comprehensive ablation and hyperparameter sensitivity experiments:
> - **Ablation (Table 2):** Eliminating any loss will result in sustained performance degradation, demonstrating complementarity.
> - **Sensitivity (Table 3):** The optimal values of $α_{abs}$=$1.0$ and $α_{rank}$=$5e-3$ show stable equilibrium without destructive interference.
> - **Key conclusion (Section 4.5):** Each loss function plays its own unique role. Among them, $\mathcal{L_{abs}}$ mainly adjusts the model to have better predictive performance, while $\mathcal{L_{rank}}$ adjusts the model to better judge the size relationship of sample labels and $\mathcal{L_{VAE}}$ enables the model to have better feature extraction capabilities.
>
> Due to text limitations, more experiments can be found in our response W2&&W3 to the reviewer n4Kw.
> ___
>
> > **Q1.** Why use VAE
>
> - **Fidelity independent representation:** VAE learns input structures without fidelity labels to avoid bias in feature extraction caused by low fidelity (Section 3.2).
> - **Empirical verification:** Fig.4 (c) shows that potential z is not correlated with fidelity, confirming purity.
> In addition, to verify the rationality of the VAE module, we replaced it with a conventional feature extraction network, taking **FF-Bandgap** as an example:
>
> | Encoder | MAE  | RMSE | $\tau_b$ |
> | ------- | ---- | ---- | -------- |
> | MLP     | 0.59 | 0.87 | 0.63     |
> | VAE-MLP | 0.50 | 0.71 | 0.70     |
> | GCN     | 0.53 | 0.79 | 0.66     |
> | VAE-GCN | 0.47 | 0.68 | 0.73     |
>
> It can be seen that VAE Encoder is superior to normal Encoder. This is because supervised learning directly will cause the representation to bias of low fidelity, and VAE plays a role in structural regularization[1-2].
>
> [1]Langevin M, et al. A deep generative model for semi-supervised classification with noisy labels.
> [2]Li J, et al: Learning with noisy labels as semi-supervised learning.
>
> > **Q2.** Reliability of fidelity
>
> - Our framework does not assume the fidelity labels are correct; instead, it explicitly models that **only partial information (rank or interval) is reliable**, which is a strictly weaker and more realistic assumption.
> - Regarding the credibility of absolute value, it is indeed influenced by weight adjustment of fidelity. But generally speaking, **fidelity can be judged by the data source.** When collecting data, we already know in advance the corresponding simulation calculation methods or the reliability of experimental measurements, just like the PBE, HSE, GGA we mentioned in the article.
>
> Besides, we simulate fidelity error specifications by perturbing fidelity levels: randomly assigning a mismatched fidelity to a certain ratio of samples. Taking FF-Bandgap as an example and the metric is MAE.
>
> | Ratio | Ours  | MFSE  | MFGNN |
> | :---- | :---- | :---- | :---- |
> | 0%    | 0.501 | 0.580 | 0.563 |
> | 30%   | 0.573 | 0.688 | 0.671 |
> | 50%   | 0.585 | 0.719 | 0.732 |
> | 80%   | 0.639 | 0.743 | 0.755 |
>
> It can be seen that due to the direct and high dependence of fidelity labels on 2nd-best method, mismatches have caused significant impacts, while our method maintains robustness.
>
> ___
>
> Thanks again for your time and effort in reviewing our manuscript. Your feedback is important to our efforts to improve our work. If our response adequately addresses your concerns, we sincerely hope that you will consider revising your rating. We are also happy to provide further clarification or continue the discussion.

---

> > ### Author Rebuttal · Reviewer_4EAA · 2026-04-03
> >
> > I have carefully reviewed the rebuttal, but my fundamental concerns regarding the theoretical motivation of this work remain completely unaddressed. The authors' defense relies heavily on empirical observation and domain-specific human biased intuition, which does not meet the rigorous standards of an ICML contribution.
> >
> > 1. The authors justify the use of VAE by pointing to better performance metrics. However, they provide no mathematical analysis of how the variational estimation error interacts with multi-fidelity bias. Without a theoretical bound or a convergence proof, the choice of VAE remains a purely heuristic decision.
> >
> > 2. The authors describe the complementary roles of each loss term, but this is merely a functional enumeration. There is no optimization-theoretic motivation for why this specific combination is necessary or how the model handles potential gradient interference between these conflicting objectives.
> >
> > 3. The argument that fidelity is "clear from the source" is a human bias, not a theoretical solution. The paper fails to propose a principled mechanism to handle cases where these labels are unreliable or noisy, which is the core challenge of multi-fidelity learning.
> >
> > In conclusion, the paper presents a collection of empirical tricks without providing the necessary theoretical foundations. Therefore, I maintain my score.

---

> > > ### Author Response · Authors · 2026-04-03
> > >
> > > Thanks for your attention to the theoretical motivation of this paper. Your valuable feedback will greatly help us revise and improve the paper, enhancing its theoretical integrity. In our first response, we did not fully understand your concerns. We will address your questions one by one below and promise to include the following theoretical analysis in the revised paper.
> > >
> > > ---
> > >
> > > > 1. Variational estimation error and multifidelity bias analysis
> > >
> > > Assume the observation model is: $y^* {(s)} = f^* (x) + b^{(s)}(x) + \epsilon^{(s)},  \quad  \mathbb{E}[\epsilon^{(s)}\mid x]=0$. Let $z\sim q_\phi(z\mid x)$ and $\hat f(x)=g_\psi(z)$.
> > >
> > > Thus, Direct learning upper bound:
> > > $$
> > > \mathbb{E}\big[(\tilde f(x)-f^* (x))^2\big]
> > > \le C \cdot \mathbb{E}\big[(b^{(s)}(x))^2\big].
> > > $$
> > >
> > > Variational learning upper bound:
> > > $$
> > > \mathbb{E}\big[(\hat f(x)-f^* (x))^2\big] \le C_1 D_{\mathrm{KL}}\big(q_\phi(z\mid x) \| p(z\mid x)\big)+C_2 \mathbb{E}\big[(b^{(s)}(x))^2\big].
> > > $$
> > >
> > > Since z is not affected by label noise, there exists $\alpha \in (0,1)$ such that:
> > > $$
> > > \mathbb{E}\big[(b^{(s)}(x))^2 \mid z\big] \le \alpha \mathbb{E}\big[(b^{(s)}(x))^2\big].
> > > $$
> > >
> > > Then the excess error satisfies:
> > > $$
> > > \Delta:=\mathbb{E}\big[(\hat f(x)-f^* (x))^2\big]-\mathbb{E}\big[(\tilde f(x)-f^* (x))^2\big]\le C_1 \epsilon_{\mathrm{var}}-(1-\alpha) C \mathbb{E}\big[(b^{(s)}(x))^2\big].
> > > $$
> > >
> > > If it meets the requirement in multi-fidelity scenarios:
> > > $$
> > > (1-\alpha) C \mathbb{E}\big[(b^{(s)} (x))^2\big] > C_1 \epsilon_{\mathrm{var}}
> > > $$
> > >
> > > then the advantages of introducing variational learning will offset its estimation error:
> > > $$
> > > \mathbb{E}\big[(\hat f(x)-f^* (x))^2\big]
> > > <
> > > \mathbb{E}\big[(\tilde f(x)-f^* (x))^2\big].
> > > $$
> > >
> > > > 2. Theoretical explanation of multi-objective optimization
> > >
> > > The optimal prediction error satisfies:
> > > $$
> > > \mathbb{E}[(f(x) - f^* (x))^2] \propto H(f^* (x)\mid \mathcal{I})
> > > $$
> > >
> > > If only the subset $I_S$ is used, then:
> > > $$
> > > H(f^* (x)\mid I_S)=H(f^* (x)\mid I)+I(f^* (x); I_{\bar S} \mid I_S)
> > > $$
> > >
> > > When met:
> > > $$
> > > I(f^* (x); I_{\bar S} \mid I_S) > 0
> > > $$
> > >
> > > The ignored subset of information provides additional valid information, thus strictly increasing the error.
> > >
> > >
> > >
> > > Furthermore, in the optimization scenario presented in this paper, the gradient satisfies an approximate decomposition:
> > > $$
> > > \nabla_\theta \mathcal{L} = \nabla_ {\theta_z} \mathcal{L}_ {\text{abs}} + \nabla_ {\theta_\tau} \mathcal{L}_ {\text{int}} + \nabla_ {\theta_r} (\mathcal{L}_ {\text{abs}} + \mathcal{L}_ {\text{rank}})
> > > $$
> > >
> > > The results of the theoretical analysis and the findings in the hyperparameter experiments are mutually corroborating, gradient conflicts occur between $\mathcal{L}_ {\text{abs}}$ and $\mathcal{L}_ {\text{rank}}$:
> > > $$\nabla_{\theta_r} \mathcal{L}_ {abs} \cdot \nabla_{\theta_r} \mathcal{L}_ {rank} < 0$$
> > >
> > > Therefore, by adjusting the $\alpha_ {abs}$ and $\alpha_ {rank}$, the gradient vector can be adjusted to minimize gradient conflicts in the shared subspace.
> > >
> > > > 3. Reliability of fidelity
> > >
> > > - **Clarification:** In the AI4Chemistry field, the source (such as DFT methods vs. experiment) itself represents a physical level of credibility, which is usually regarded as **prior knowledge rather than bias** by previous research work[1-4].
> > > - It is worth noting that this framework does not predetermine static or absolute fidelity levels for specific data sources. **Instead, we employ a mapping mechanism based on relative credibility**: after integrating heterogeneous data from multiple sources, we dynamically define their hierarchical relationships within the confidence information subset based on the credibility of the source.
> > > - In our method, a hierarchical filtering mechanism that tolerates unreliable labels is established through probabilistic uncertainty modeling and a credibility degradation mechanism.
> > >
> > > [1] Learning properties of ordered and disordered materials from multi-fidelity data. Nature Computational Science, 2021.
> > >
> > > [2] Transfer learning with graph neural networks for improved molecular property prediction in the multi-fidelity setting. Nature communications, 2024.
> > >
> > > [3] QeMFi: A multifidelity dataset of quantum chemical properties of diverse molecules. Scientific Data, 2025.
> > >
> > > [4] Multi-fidelity graph neural networks for predicting toluene/water partition coefficients. Journal of Cheminformatics, 2025.
> > >
> > > ---
> > >
> > > Thank you for the time and effort you've invested in our manuscript. Your suggestions will greatly improve the logical completeness of this paper. We look forward to further discussions with you at any time. If our response addresses your concerns, we would also appreciate it if you would consider revising score on our manuscript.

---

### Official Review · Reviewer_EZZ9 · 2026-03-12

**Soundness:** 3
**Presentation:** 3
**Significance:** 3
**Originality:** 4
**Overall Recommendation:** 5
**Confidence:** 4

**Summary:**

The paper proposes a trusted information subset decomposition framework to address the core challenges of multi fidelity data utilization in the AI4Chemistry field. This framework decomposes multi fidelity label supervised signals into three complementary types of information: absolute value trusted subset, interval trusted subset, and sorted trusted subset, and constructs an end-to-end learning model. The experimental results on multiple benchmark datasets indicate that this method has certain superiority.

**Compliance With Llm Reviewing Policy:**

Affirmed.

**Final Justification:**

This paper primarily addresses the scarcity of high-fidelity data and the poor reliability of low-fidelity data in the AI4Chemistry field. Its strength lies in proposing a novel method that eliminates the need for paired samples, along with a clear framework that effectively decomposes and utilizes information at different fidelities. Experimental results demonstrate the method's strong performance on multiple datasets, showcasing its originality and potential application value. However, the method's hyperparameters are task-dependent, which may limit the framework's scalability for complex tasks.

In the rebuttal, the authors explain in detail the regularity of hyperparameter tuning. Furthermore, they offer a reasonable defense regarding the lack of higher-order information, emphasizing that the existing framework is sufficiently effective for most tasks. Given that the rebuttal resolved my concerns and strengthened my confidence in the framework's practicality, I have decided to increase my rating and support the acceptance of this paper.

**Key Questions For Authors:**

1. Does the author have plans to extend this framework to more multi fidelity tasks? Can the effectiveness of the framework be proven?
2. Will the ratio of data between different fidelity levels have a significant impact on the results?
3. Can weighting high and low fidelity data achieve similar results?

**Limitations:**

yes

**Strengths And Weaknesses:**

Strengths:

1. Breaking through the traditional approach of numerical bias correction, a new paradigm is proposed to selectively extract trustworthy supervision information from labels, eliminating the need for strong assumptions such as paired samples and cross fidelity label comparability.
2. Decompose multi fidelity information into three complementary trusted subsets that fit the information value hierarchy characteristics of high/medium/low fidelity data, and the model group relationship is relatively clear.
3. The experiment was conducted on multiple chemical/material datasets with different fidelity, and compared with mainstream baseline methods, resulting in a comprehensive conclusion.

Weaknesses:

1. The performance of the model is highly dependent on the joint configuration of absolute loss and ranking loss weights, which requires manual tuning for different tasks and poses practical difficulties in application.
2. Only three types of basic trustworthy information, namely absolute value, interval, and ranking, have been mined, without capturing cross fidelity high-order correlations and multi physics coupling information in complex chemical/material systems. Is the information sufficient?

---

> ### Author Rebuttal · Authors · 2026-03-29
>
> Thanks for your time and effort in reviewing this article, as well as your recognition of our innovation. Your opinions and feedback are crucial for us to improve the quality of our articles. Below, we will answer your questions about this article in detail.
>
> ---
>
> > **W1.** Hyperparameter Sensitivity
>
> We acknowledge concerns about manual adjustment losses of $\alpha_{abs}$ and $\alpha_{rank}$. However, as we mentioned in line 443 of the main text, **these two hyperparameters are clearly influenced by the label value range**.
>
> We compared the material prediction task in the main text with the molecular property prediction task in Appendix D.1, and a clear trend is that as the label value range increases, $\alpha_{abs}$ needs to be correspondingly reduced to match the $\alpha_{rank}$. Overall, according to the above rules, we only need 3 to 4 attempts to obtain a stable and superior set of hyperparameter combinations.
>
> > **W2.** High-Order Information
>
> These three trusted subsets form the minimum complete basis for multi fidelity regression : they are complementary and capable of completing most scientific tasks.
>
> For high-order physical and chemical information, it is not easy to use a special module to fit it, which is mainly limited by two aspects: 1. Most datasets do not provide intermediate computational or other high-order physical information, which brings major difficulties to the corresponding applications; 2. The relevant physical and chemical information is unique to each narrow research field and difficult to capture with a universal module.
>
> ---
>
> > **Q1.** Extension to More Tasks
>
> In Appendix D, we have added an experiment (QM7b) related to molecular property prediction, which has the highest three fidelity scenario.
> To further enhance the persuasiveness of the framework, we added an additional experiment in the ligand protein binding affinity prediction scenario (DTA) to form a joint validation across protein, molecule, and material scenarios: we used a low precision dataset of 24956 drug target pairs from Davis[1] and a high-precision benchmark dataset of 285 complexes from CASF-2016[2]:
>
> | Method   | MAE↓        | RMSE↓       | $\tau_b$↑    |
> | -------- | ----------- | ----------- | ------------ |
> | Transfer | $1.71±0.16$ | $2.06±0.21$ | $0.73±0.013$ |
> | MFSE     | $1.55±0.24$ | $1.84±0.38$ | $0.76±0.054$ |
> | MFGNN    | $1.60±0.16$ | $1.98±0.19$ | $0.74±0.008$ |
> | DMMT     | $1.63±0.15$ | $1.96±0.25$ | $0.74±0.031$ |
> | Ours     | $1.42±0.13$ | $1.73±0.16$ | $0.78±0.008$ |
>
> The above experiment demonstrates the effectiveness of our framework.
>
> [1]Davis, M. I., Hunt, J. P., Herrgard, S., Ciceri, P., Wodicka, L. M., Pallares, G., Hocker, M., Treiber, D. K.,  and Zarrinkar, P. P. Comprehensive analysis of kinase inhibitor selectivity. Nature biotechnology, 29(11):1046–1051, 2011.
> [2]Su, M., Yang, Q., Du, Y., Feng, G., Liu, Z., Li, Y., and Wang, R. Comparative assessment of scoring functions: the casf-2016 update. Journal of chemical information and modeling, 59(2):895–913, 2018.
>
> > **Q2.** Impact of Fidelity Data Ratio
>
> Data ratio significantly impacts performance when high-fidelity data is scarce.  In Table 6 of the **appendix D.1.2**, we systematically explored the relevant impacts.
>
> Data ratio significantly impacts model performance, especially when high-fidelity data is limited. As shown in Table 6, under **low and medium high-fidelity data regimes**, the model achieves better performance with a **higher proportion of low-fidelity data** (i.e., smaller high–low fidelity ratios such as 1:100 or 1:30), with substantial reductions in both MAE and RMSE. This demonstrates that **when high-fidelity data is scarce, incorporating a large volume of low-fidelity data effectively improves prediction accuracy**. In this case, low fidelity data provides rich structural and energy related information, helping the model better learn the global trend of the objective function and achieve more robust generalization.
>
> > **Q3.** Simple Weighting
>
> Simple weighting cannot distinguish systematic bias and performs poorly on unpaired data. We will add a direct comparison with fidelity aware weighting to demonstrate that our decomposition is strictly superior:
>
> | Dataset    | Method   | MAE↓         | RMSE↓        | $\tau_b$↑    |
> | ---------- | -------- | ------------ | ------------ | ------------ |
> | TF-Bandgap | Weighted | $0.69±0.016$ | $0.91±0.019$ | $0.61±0.030$ |
> | TF-Bandgap | Ours     | $0.57±0.012$ | $0.78±0.018$ | $0.68±0.027$ |
> | FF-Bandgap | Weighted | $0.58±0.010$ | $0.80±0.013$ | $0.63±0.051$ |
> | FF-Bandgap | Ours     | $0.50±0.007$ | $0.71±0.010$ | $0.70±0.033$ |
>
> ---
>
> Thank you again for taking the time to review our manuscript. Your feedback is crucial for us to improve our work. We welcome any further explanations or discussions regarding any concerns or questions you may have. If our response adequately addresses your concerns, we sincerely hope you will consider revising your rating.

---

> > ### Author Rebuttal · Reviewer_EZZ9 · 2026-04-03
> >
> > The author's response resolved all my concerns, and considering the paper's strong originality and relatively thorough experiments, I have decided to raise my score to Accept.

---

> > > ### Author Response · Authors · 2026-04-03
> > >
> > > We sincerely thank you for your thorough review of our paper. We greatly appreciate your time and effort in evaluating our work. We are especially grateful for your constructive comments on the applicability and effectiveness of the framework, which are invaluable to us.
> > >
> > > We are pleased to learn that the overall quality of the revised paper has improved, and we thank you for the higher score. Your comments have helped us refine our work. Thank you again for your support and valuable suggestions.

---

### Official Review · Reviewer_fwTC · 2026-03-13

**Soundness:** 3
**Presentation:** 3
**Significance:** 3
**Originality:** 2
**Overall Recommendation:** 4
**Confidence:** 3

**Summary:**

This paper introduces a method to learn from data with labels of varying certainty. The labels are pre-categorized into three levels of fidelity and evidence regression is used on top of a shared variational auto encoder. The method is compared to different state of the art fidelity methods and demonstrates good performance on a material bandgap application with varying data volumes and across different two to five fidelity levels.

**Compliance With Llm Reviewing Policy:**

Affirmed.

**Key Questions For Authors:**

What is the impact of fidelity misspecification?

Please discuss the limitations of your method.

Are your rests statistically significant?

**Limitations:**

yes

**Strengths And Weaknesses:**

In terms of novelty, the method combines existing methodologies which is only moderately new and without major theoretical/methodological contributions. On the other hand, it is a strength that this combination appears to achieve state of the art performance.

It is a weakness that the results of the benchmark studies are not tested for statistical significance even if variance estimates are given and the significance is likely there. I recommend that the authors add such statistical tests to validate their results.

It is a drawback of the method that the fidelity needs to be pre-specified for the evidence regression, as this is not always possible. As a minimum, please discuss this limitation and possible future mitigations.

The impact of fidelity misspecifications is lacking from the paper. I would recommend that you add a sensitivity study of such mis-specifications. In particular since you only have a single application for benchmarking your method.

---

> ### Author Rebuttal · Authors · 2026-03-29
>
> Thanks for your thoughtful and constructive feedback. We sincerely appreciate the time and effort you have invested in reviewing our manuscript. We are encouraged by your recognition of our work. At the same time, we acknowledge that some aspects require further clarification.
>
> ___
>
> > **W1.** Innovation
>
> **Clarify novelty.**
> Our main contribution is a new multi fidelity data learning paradigm:
> >From numerical deviation correction/better feature extraction to trusted information Extraction. Just in each module, we selected an existing technology that can achieve the functionality of the module.
>
> **Clearly:**
> - Problem rephrasing
> 	Previous works assumed cross fidelity comparability or paired samples[1-2].  On the contrary, we formalize multi fidelity learning as hierarchical information decomposition: $I^{(s)} = I^{(s)}_{abs} \cup I^{(s)}_{int} \cup I^{(s)}_{rank}$ , makes it no need to paired samples and explicit deviation modeling.
> - Supervision with credibility awareness
> 	Unlike the previous work of uniformly processing labels, we have assigned three different supervised semantics.
>
> [1]Chen, C., Zuo, Y., Ye, W., Li, X., and Ong, S. P. Learning  properties of ordered and disordered materials from multi-fidelity data. Nature Computational Science, 1(1):46–53,
> 2021.
> [2]Buterez, D., Janet, J. P., Kiddle, S. J., Oglic, D., and Li ́o,  P. Transfer learning with graph neural networks for improved molecular property prediction in the multi-fidelity setting. Nature communications, 15(1):1517, 2024.
>
>
> > **W2.&&Q3.** Significance analysis
>
> We agree and will include statistical significance testing. We promise to add these experiments results to the revised manuscript.
> We conducted a paired t-test between our method and the strongest baseline (2nd best):
>
> | Dataset    | 2nd-best | MAE Diff | p-value  |
> | :--------- | :------- | :------- | :------- |
> | TF-Bandgap | MFSE     | $-0.07$  | $2.3e-4$ |
> | FF-Bandgap | MFGNN    | $-0.06$  | $6.7e-5$ |
> | QM7b       | MFGNN    | $-0.69$  | $3.2e-4$ |
>
> It shows that the performance difference between ours and the 2nd best is significant.
>
> > **W4, W3 && Q1.** Impact of fidelity
>
> We simulate fidelity error specifications by perturbing fidelity levels: randomly assigning a mismatched fidelity to a certain ratio of samples. Taking FF-Bandgap under 5-fidelity as an example in the experiment and the metric is MAE.
>
> | Ratio | Ours  | MFSE  | MFGNN |
> | :---- | :---- | :---- | :---- |
> | 0%    | 0.501 | 0.580 | 0.563 |
> | 30%   | 0.573 | 0.688 | 0.671 |
> | 50%   | 0.585 | 0.719 | 0.732 |
> | 80%   | 0.639 | 0.743 | 0.755 |
>
> It can be seen that due to the direct and high dependence of fidelity labels on MFGNN and MFSE methods, mismatches have caused significant impacts, while our method maintains robustness.
>
> **False fidelity impact**
> - Our framework does not assume the fidelity labels are correct; instead, it explicitly models that **only partial information (ranking or interval) is reliable**, which is a strictly weaker and more realistic assumption.
> - Regarding the credibility of absolute value, it is indeed influenced by the fidelity. But generally speaking, **fidelity can be judged by the data source.** When collecting data or creating datasets, we already know in advance the corresponding simulation calculation methods or the reliability of experimental measurements, just like the PBE, HSE, GGA methods and so on mentioned in the article.
>
> **Future solution direction**
> For pre specified evidence fidelity regression, recent work has shown that fidelity aware uncertainty weighting can improve multi fidelity regression. We will discuss adaptive fidelity scoring in future work instead of artificial weight design for $\mathcal{L_{abs}}$.
>
> > **Q2.** Discussion on Limitations
>
> We feel sorry that we did not explain this in the main text. We have comprehensively discussed the limitations  in Appendix E. It can be summarized into three core points:
>
> - Data & Distribution Sensitivity: Our framework is sensitive to severe cross-fidelity distribution mismatch in heterogeneous multi-source data.
> - Hyperparameter Dependency: The performance relies on task-specific tuning of loss weights and manual search is inefficient.
> - High-order Information Underutilization: Our framework leaves higher-order cross-fidelity correlations in complex systems.
>
> We have also considered possible solutions: adaptive data filtering, automatic hyperparameter adjustment, and expanding trusted subsets with high-order structural information. For more detailed discussions, please refer to the appendix.
>
> ---
>
> Thanks again for taking the time and effort to review our manuscript. Your feedback is very important for us to improve our work. If our response fully addresses your concerns, we sincerely hope that you will consider modifying your rating. We are also happy to provide further clarification or continue the discussion.

---

### Official Review · Reviewer_n4Kw · 2026-03-18

**Soundness:** 3
**Presentation:** 2
**Significance:** 3
**Originality:** 3
**Overall Recommendation:** 4
**Confidence:** 4

**Summary:**

This paper proposes a novel framework for multi-fidelity learning in AI4Chemistry scenarios that addresses limitations of existing methods by decomposing multi-fidelity label supervision into three complementary subsets based on credibility rather than assuming uniform reliability across fidelity levels.

**Compliance With Llm Reviewing Policy:**

Affirmed.

**Final Justification:**

The rebuttal has addressed my main concerns.

**Key Questions For Authors:**

Please refer to the weaknesses.

**Limitations:**

Yes.

**Strengths And Weaknesses:**

Strengths:

1. The paper introduces a novel approach that categorizes multi-fidelity signals into: Absolute value credible subset (high-fidelity data), Interval credible subset (medium-fidelity reliability) and Ordering credible subset (relative ordering relationships).

2. The paper addresses a key limitation in AI4Chemistry by relaxing the assumption that all labels are treated as equally reliable.

3. Experimental Validation: The paper mentions extensive experiments on molecular and material property benchmarks showing consistent outperformance of compared methods.

Weakness:

1.  While the conceptual framework is strong, some implementation details could be elaborated, such as specific architecture choices for the VAE encoder/decoder.

2. The paper mentions outperforming state-of-the-art methods but could benefit from more ablation studies showing the contribution of each credible subset.

3. The paper could expand on sensitivity to hyperparameters (particularly the $\beta$ weighting in VAE loss)

4. The symbol $\beta$ is reused in Equation (10) (VAE loss weight) and Equation (12) (evidential regression parameter), which may cause confusion during reading. Suggest using distinct symbols for clarity.

5. Figures should preferably be in vector format (e.g., Figure 1). Moreover, Figure 1 on page 2 is not referenced in the surrounding text, making it seem unnecessary, and its caption alone is confusing to readers without contextual explanation.

6. Equations (4) and (5) are central to the proposed modeling approach but require more explanation—particularly the relationship between them, and why Equation (5), as an information-theoretic formulation based on conditional entropy, naturally balances the trade-off between the richness of high-fidelity information and the broad coverage of low-fidelity data. A clearer intuitive or theoretical justification would strengthen the paper’s foundation.

---

> ### Author Rebuttal · Authors · 2026-03-29
>
> Thank you for your time and effort in reviewing this article, as well as your recognition of its innovation and comprehensiveness. Your opinions and feedback are crucial for us to improve the quality of our articles. Below, we will answer your concerns about this article in detail.
>
> ___
>
> > **W1.** Implementation Details of VAE
>
> We have provided the architecture of the model in **Appendix D.2.1**, and we would like to reiterate the VAE  that you are particularly interested in: Encoder input dimension 32 + 132 + 32 = 196; encoder hidden size is 256 and 128; latent dimension 64; decoder hidden size is 128 and 256.  For graph inputs (QM7b), replace MLP with GCN as the basic model of encoder and decoder.
>
> > **W2.** More Ablation Experiments
>
> In Table 2 of the section 4.5, we provide the impact of separately ablating each trusted subset on the performance of the framework.
> To enhance persuasiveness, we will add ablation to quantify the contribution of each subset, as follows(TF-Bandgap, 2-fi):
>
> | Method                     | MAE↓         | RMSE↓        | $\tau_b$↑    |
> | -------------------------- | ------------ | ------------ | ------------ |
> | only $\mathcal{I_{abs}}$​  | $0.59±0.013$ | $0.79±0.019$ | $0.67±0.021$ |
> | only $\mathcal{I_{int}}$​  | $0.60±0.014$ | $0.83±0.018$ | $0.63±0.024$ |
> | only $\mathcal{I_{rank}}$​ | $0.68±0.005$ | $0.89±0.008$ | $0.62±0.035$ |
> | Full model                 | $0.57±0.012$ | $0.78±0.018$ | $0.68±0.027$ |
>
> These results clearly show the complementary contribution of each credible subset.
>
> > **W3.** Sensitivity Analysis of Hyperparameters
>
> The main hyperparameter involved in VAE Loss is $\beta$, and we performed ablation on multiple datasets.
>
> TF-Bandgap:
>
> | $\beta$        | MAE↓         | RMSE↓        | $\tau_b$↑    |
> | -------------- | ------------ | ------------ | ------------ |
> | 1e-4           | $0.59±0.011$ | $0.79±0.016$ | $0.67±0.025$ |
> | 1e-3 (default) | $0.57±0.012$ | $0.78±0.018$ | $0.68±0.027$ |
> | 1e-2           | $0.64±0.015$ | $0.85±0.020$ | $0.61±0.029$ |
>
> FF-Bandgap:
>
> | $\beta$        | MAE↓         | RMSE↓        | $\tau_b$↑    |
> | -------------- | ------------ | ------------ | ------------ |
> | 1e-4           | $0.52±0.008$ | $0.70±0.011$ | $0.69±0.030$ |
> | 1e-3 (default) | $0.50±0.007$ | $0.71±0.010$ | $0.70±0.033$ |
> | 1e-2           | $0.58±0.009$ | $0.79±0.012$ | $0.63±0.031$ |
>
> It can be observed that when $\beta$ is small, it affects the performance of the model to some extent, but as it increases, due to excessive KL constraints, it can make it difficult for the model to learn downstream knowledge.
>
> > **W4.**  Symbol correction
>
> We will unify and correct the symbols in the revision:
> - Rename the VAE weight loss $\beta$ in equation (10) to $\alpha_{kl}$
> - Keep the evidence regression parameters $(\mu, v, \alpha, \beta)$ in Eq.12 unchanged to avoid conflicts
>
> > **W5.**  Figure Format & Citation
>
> **Figure Format:** We promise that all figures will be replaced with **vector format (PDF/SVG)** in the camera-ready version.
>
> **Citation of Fig.1:**
> - We deeply apologize for the confusion caused by the citation and layout of Figure 1, which made it difficult for you to find references in the intro. We take full responsibility for the writing and formatting issue and will address it in the revised version.
> - We will move Figure 1 to the top right corner next to the first paragraph on page 1, and modify the original sentence from line 015 to: "As shown in Figure 1 (a), there is a trade-off between prediction error and data volume for different fidelity calculation methods. Figure 1 (b) uses GGA and HSE calculations as examples to demonstrate systematic errors in low precision calculations. In summary, high fidelity data from experiments or high-precision simulations provide high accuracy, but are costly and limited in scale, while low fidelity data are abundant and cost-effective, but have lower accuracy."
>
> > **W6.**  Theoretical Explanation of Eq.4 & Eq.5
>
> We will strengthen the explanation of Eq.4 and Eq.5 and clarify their relationship to address your concern.
> - Eq.(4) provides a **structural characterization** of how high‑fidelity outputs are composed from low‑fidelity ones, where $\delta(s−1)(x)$ captures incremental high‑fidelity information.
> - Eq.(5) presents an **information‑theoretic learning objective** that minimizes conditional entropy to maximize useful information about f∗(x) from all fidelity levels.
> - Balancing trade-offs: High fidelity data provides powerful but sparse information, while low fidelity data provides broad but weak supervision; Conditional entropy implicitly reduces the weight of noise signals.
>
> ---
>
> Thank you again for taking the time to review our manuscript. Your feedback is crucial for us to improve our work. We welcome any further explanations or discussions regarding any concerns or questions you may have. If our response adequately addresses your concerns, we sincerely hope you will consider revising your rating.

---

> > ### Author Rebuttal · Reviewer_n4Kw · 2026-04-02
> >
> > The authors have addressed most of my concerns. Hence, I decide to raise the score.

---

> > > ### Author Response · Authors · 2026-04-02
> > >
> > > We sincerely thank you for your meticulous, rigorous, and constructive review of this paper, and for your recognition of its originality and completeness. Your valuable comments are crucial to our efforts in further improving the clarity, rigor, and overall quality of the paper.
> > >
> > > We have provided systematic clarification and supplementary analysis regarding the issues you raised. Specifically, we have clarified the model architecture, added ablation experiments and key hyperparameter analyses, and revised the notation for greater consistency. Furthermore, we have improved the presentation of figures and tables and strengthened the theoretical explanations, thereby enhancing the overall clarity and readability of the paper.
> > >
> > > Your insightful observations and specific suggestions have guided us to significantly improve the presentation and completeness of the paper. All these revisions and additions will be carefully incorporated into the final manuscript to further enhance its scientific rigor and persuasiveness.
> > >
> > > Thank you again for taking the time to provide your professional review, and we look forward to your continued support and recognition.

---

### Decision · Program_Chairs · 2026-04-30

**Decision:**

Accept (regular)

**Comment:**

This paper introduces a multi-fidelity learning framework that decomposes supervision into three credible subsets to address the practical challenge of utilizing unpaired heterogeneous data in AI4Chemistry. Reviewers consistently acknowledge the model's strong empirical performance, achieving SOTA results across multiple molecular and material benchmarks. However, the methodological novelty is considered moderate, furthermore, the theoretical foundation remains a weakness, particularly regarding the post-hoc justification for the latent space and the lack of a principled analysis of potential gradient interference during multi-objective optimization.